

# 1 Influence of Ocean Alkalinity Enhancement with Olivine or Steel
# 2 Slag on a Coastal Plankton Community in Tasmania

Jiaying A. Guo[1,2], Robert F. Strzepek[2], Kerrie M. Swadling[1,2], Ashley T. Townsend[3], Lennart T. Bach[1]
[1]Institute for Marine and Antarctic Studies, University of Tasmania, Hobart, Tasmania, 7000 Australia
[2]Australian Antarctic Program Partnership (AAPP), Institute for Marine and Antarctic Studies, University of Tasmania,
Hobart, Tasmania, 7000 Australia
[3]Central Science Laboratory, University of Tasmania, Sandy Bay, Tasmania, 7005 Australia
*Correspondence to*: Jiaying A. Guo (Jiaying.guo@utas.edu.au)
**Abstract.** Ocean alkalinity enhancement (OAE) aims to increase atmospheric $CO_2$ sequestration in the oceans through the
acceleration of chemical rock weathering. This could be achieved by grinding rocks containing alkaline minerals and
adding the rock powder to the surface ocean where it dissolves and chemically locks $CO_2$ in seawater as bicarbonate.
However, $CO_2$ sequestration during dissolution coincides with the release of potentially bio-active chemicals and may
induce side effects. Here, we used 53 L microcosms to test how coastal plankton communities from Tasmania respond to
OAE with olivine (mainly $Mg_2SiO_4$) or steel slag (mainly $CaO$ and $Ca(OH)_2$) as alkalinity sources. Three microcosms were
left unperturbed and served as a control, three were enriched with olivine powder (1.9 g $L^{-1}$), and three with steel slag
powder (0.038 g $L^{-1}$). Phytoplankton and zooplankton community responses as well as some biogeochemical parameters
were monitored for 21 days. Olivine and steel slag additions increased total alkalinity by 29 µmol $kg^{-1}$ and 361 µmol $kg^{-1}$
respectively, which corresponds to a theoretical increase of 0.9 % and 14.8 % of the seawater storage capacity for
atmospheric $CO_2$. Olivine and steel slag released silicate nutrients into the water column, but steel slag released
considerably more and also significant amounts of phosphate. Both minerals released dissolved aluminium (>400 nmol $L^{-1}$).
The slag addition increased dissolved manganese concentrations (784 nmol $L^{-1}$), while olivine increased dissolved nickel
concentrations (38 nmol $L^{-1}$). The slag treatment increased the total particulate manganese concentrations (22 nmol $L^{-1}$),
while olivine increased the total particulate nickel (5 nmol $L^{-1}$), which was consistent with the increase in the dissolved
concentrations of these trace metals in seawater. There was no significant difference in total chlorophyll *a* concentrations
between the treatments and the control, likely due to nitrogen limitation of the phytoplankton community. However, flow
cytometry results indicated an increase in the cellular abundance of several smaller (~<20 µm) phytoplankton groups in
the olivine treatment compared to the slag treatment and the control. The abundance of larger phytoplankton (~>20 µm)
decreased much more in the control than in the mineral addition treatments after day 10. Furthermore, the maximum
quantum yields of photosystem II ($F_v/F_m$) were higher in slag and olivine treatments, suggesting that mineral additions
increased photosynthetic performance. The zooplankton community composition was also affected with the most notable
changes being observed in the dinoflagellate *Noctiluca scintillans* and the appendicularian *Oikopleura* sp. Overall, steel
slag is much more efficient for $CO_2$ removal with OAE than olivine and appears to be induce less changes in the plankton
community when relating the $CO_2$ removal potential to the level of environmental impact that was observed here.





## 1 Introduction

Keeping global warming below 2 °C requires immediate emissions reduction. Additionally, between 450-1100 Gigatonnes of carbon dioxide ($CO_2$) need to be removed from the atmosphere by 2100 (Smith et al., 2023). This could be achieved with a portfolio of terrestrial and marine Carbon Dioxide Removal (CDR) methods. Ocean alkalinity enhancement (OAE) is a marine CDR method that could theoretically contribute significantly to the global CDR portfolio (Ilyina et al., 2013; Feng et al., 2017; Lenton et al., 2018).

Alkalinity is generated naturally when rock weathers and it has control on the ocean's chemical capacity to store $CO_2$ (Schuiling and Krijgsman, 2006). Natural rock weathering is currently responsible for about 0.5 Gt of atmospheric $CO_2$ sequestration every year (Renforth and Henderson, 2017). The idea behind OAE is to accelerate natural rock weathering by extracting calcium- or magnesium-rich rocks, such as olivine, pulverizing them, and spreading them onto the sea surface to increase chemical weathering rates (Hartmann et al., 2013). The weathering (i.e., dissolution) of these alkaline minerals will consume protons ($H^+$), which shifts the carbonate chemistry equilibrium in seawater from $CO_2$ towards increasing bicarbonate ($HCO_3^-$) and carbonate ion ($CO_3^{2-}$) concentrations:

$$CO_2 + H_2O \rightleftharpoons H_2CO_3 \rightleftharpoons HCO_3^- + H^+ \rightleftharpoons CO_3^{2-} + 2H^+ \qquad (1)$$

thereby making new space for atmospheric $CO_2$ to be dissolved in seawater and permanently stored. Previous model studies have shown that OAE can mitigate climate change significantly by increasing the oceanic uptake of $CO_2$ from the atmosphere (Kohler et al., 2010; Paquay and Zeebe, 2013; Keller et al., 2014; Lenton et al., 2018). For example, the study by Burt et al. (2021) suggested that the total global mean dissolved inorganic carbon (DIC) inventories would increase by 156 GtC after total alkalinity is enhanced at a rate of 0.25 Pmol year$^{-1}$ in 75-year simulations.

There are a variety of alkaline minerals that could be used for OAE. A widely considered naturally occurring mineral is forsterite, a ($Mg_2SiO_4$)-rich olivine. This type of olivine is abundant in ultramafic rock such as dunite, constituting at least 88 % of the rock composition (Ackerman et al., 2009; Su et al., 2016). Olivine occurs in the Earth's crust but is more abundant in the upper mantle. There are at least several billion tons of olivine resources on Earth (Caserini et al., 2022). However, the extraction of olivine in 2017 was only around 8.4 Mt year$^{-1}$ (Reichl et al., 2018), which is about two orders of magnitude below the mass needed for climate-relevant OAE with olivine (Caserini et al., 2022). The net reaction for $CO_2$ sequestration with $Mg_2SiO_4$ is:

$$Mg_2SiO_4 + 4CO_2 + 4H_2O \rightarrow 2Mg^{2+} + 4HCO_3^- + H_4SiO_4 \qquad (2)$$

Another potential OAE source material is steel slag (Renforth, 2019), a by-product of steel manufacturing. During steel manufacturing, high-purity calcium oxide (CaO) is used to improve the quality of the steel through accumulation of unwanted materials such as sulphur and phosphorus. Steel slag mainly contains CaO, $SiO_2$, $Al_2O_3$, $Fe_2O_3$, MgO, and MnO (Kourounis et al., 2007), and the chemical composition can vary depending on the manufacturing process (Wang et al., 2011). Due to the presence of CaO and potentially other alkaline components, steel slag can increase alkalinity when dissolved in seawater. The chemical reaction for $CO_2$ sequestration with CaO is:



$CaO + H_2O \rightarrow Ca(OH)_2$ and $Ca(OH)_2 + 2CO_2 \rightarrow Ca^{2+} + 2HCO^-$ (3)

Some of the steel slag that is produced during steel manufacturing is further used (e.g. for road construction and civil
engineering) but in some countries like China, 70.5 % of steel slag is left unused and stored in dumps (Guo et al., 2018).
In 2016, more than 300 million tons of steel slag was not used effectively, thereby occupying the land and raising
environmental concerns (Guo et al., 2018). The effective alkaline composition, availability, and relatively low cost of the
raw materials make olivine and steel slag potential source materials for OAE.

To assess whether OAE is viable, it needs to be understood how its application may affect marine biota such as plankton
and the biogeochemical fluxes they drive. Some data on the effects of OAE with sodium hydroxide (NaOH) on plankton
communities have recently been published (Ferderer et al., 2022; Subhas et al., 2022), but to the best of our knowledge, no
such data is available for olivine- and/or slag-based OAE. Chemical perturbations via olivine and slag should be like those
by NaOH in that they increase seawater pH and shift the carbonate chemistry equilibrium (see Eq. 1). However, there
would be additional chemical perturbations because minerals contain a variety of potentially bioactive elements that are
released into the environment when they dissolve in seawater (Bach et al., 2019). One particular concern is that natural and
anthropogenic minerals such as olivine and steel slag are rich in bioactive metals that are usually scarce in the ocean, such
as iron (Fe), copper (Cu), nickel (Ni), manganese (Mn), zinc (Zn), cadmium (Cd), and chromium (Cr). Many of these trace
metals are essential micronutrients for phytoplankton growth (Sunda, 2000; Sunda, 2012), such as being co-factors for
various metalloenzymes (summarized by Twining and Baines (2013)). It is possible that the addition of alkaline minerals
may benefit phytoplankton by providing trace metals currently limiting phytoplankton growth (Falkowski, 1994; Basu and
Mackey, 2018). For instance, the addition of Fe is well known to stimulate phytoplankton blooms in those vast ocean
regions where Fe levels limit growth (Boyd et al., 2007; Moore et al., 2013). However, some trace metals can also inhibit
phytoplankton growth, and different phytoplankton species have different requirements and tolerances for trace metals
(Sunda, 2012) so the addition of trace metals via OAE may change the plankton community composition.

Here, we describe a microcosm experiment with coastal Tasmanian plankton communities that was used to investigate: (1)
how effectively OAE via the application of finely ground olivine and steel-slag could sequester atmospheric $CO_2$, and (2)
if /how olivine and steel-slag additions affect various components of the plankton community.





## 2 Methodology

### 2.1 Microcosm setup

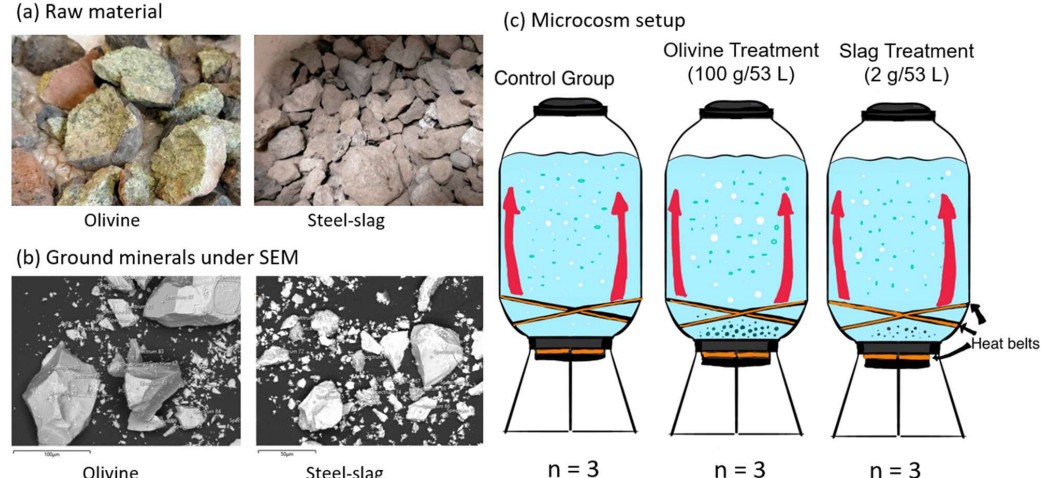

**Fig. 1.** Experimental design and alkalinity sources. (a) Raw materials used as alkalinity sources: olivine (left) and steel-slag (right). Olivine and steel-slag were originally larger than 20 mm. (b) Ground minerals observed with a scanning electron microscope. (c) Microcosm setup: each microcosm enclosed ~ 53 L of surface seawater with natural plankton communities. Olivine and steel-slag treatments and the control were kept in a temperature-controlled room and two heat belts were attached to the bottom of each microcosm to create convective circulation.

We used nine 53 L transparent Kegland® Fermzilla conical unitank fermenters (polyethylene terephthalate) (Fig. 1) as microcosms to incubate natural plankton communities. All microcosms were prewashed with hydrochloric acid (10 % v/v) and rinsed five times with 18.2 MΩ Milli-Q water. Seawater with coastal plankton communities was collected at Battery Point, Tasmania (42.892°S, 147.337°E) within 2 hours by lowering the microcosms into the ocean with a crane and filling them in a manner similar to a Niskin bottle, as described in detail in Ferderer et al. (2022). A sieve with a mesh size of 2 mm was attached to the top and bottom of the microcosms during filling to avoid the entrapment of large and patchily distributed organisms in the microcosms. The enclosed seawater weight was initially between 52.35-54.70 kg. After seawater collection, filled microcosms were immediately transported back to the Institute for Marine and Antarctic Studies (University of Tasmania) on a truck and transferred within 75 min into a temperature-controlled room set to 7.5-8 °C. Two heat belts were attached to the bottom of each microcosm to induce a convective mixing current (Ferderer et al., 2022). Seawater temperature inside the microcosms was about 13.5 °C due to the heating effects of the heat belts, and was the same as the sampled region. LED light strips were used to provide an average light intensity of 236 µmol photons m$^{-2}$ s$^{-1}$ (ranging from 208 to 267 µmol photons m$^{-2}$ s$^{-1}$) with a daily light-dark cycle of 10:14 hours. The light intensity was the average light intensity in each microcosm measured with a LICOR light meter at 0.15 m depth within the microcosm. Microcosms positioned in the temperature-controlled room were shuffled anti-clockwise every day to ensure similar light intensity for each microcosm throughout the experiment. Treatments were established 24 hours after collecting the seawater. The total alkalinity released per amount of mineral powder added was much higher for the slag powder than the olivine





powder in our preliminary test trials. So, three microcosms were enriched with 100 g of olivine powder, three microcosms
with 2 g of steel slag powder, while the remaining three microcosms were left unperturbed and served as controls.

**2.2 Preparation of olivine and steel slag powder**
The olivine rocks were provided by Moyne Shire Council who sourced the mineral from a quarry in Mortlake, Victoria,
Australia. The Basic Oxygen Slag (hereafter referred to as "slag") was provided by Bradley Mansell who sourced the
material from Liberty Primary Steel Whyalla Steelworks in Whyalla, South Australia, Australia. Upon delivery, the olivine
rocks were 40-80 mm in diameter, and slag aggregates were 20-50 mm in diameter. These were crushed to smaller than 10
mm pieces using a hydraulic crusher. The crushed material was further ground with a ring mill with a chrome milling pot.
Afterwards, finely-ground samples were sieved to get samples with $150 \sim 250$ µm grain size. The sieved olivine and slag
grains were inspected for their appearance and elemental composition using a Hitachi SU-70 analytical field emission
scanning electron microscope (SEM), and energy dispersive spectrometers (Central Science Laboratory (CSL), University
of Tasmania). Grain size spectra were determined with a Sympatec QICPIC particle size analyser LIXCELL (CSL,
University of Tasmania).

**2.3 Seawater sampling**
Seawater was transferred with a peristaltic pump from the microcosms at a depth of about 0.15 m into 1 L acid-washed
sampling bottles (LDPE) using an acid-washed silicon tube. Seawater in these bottles was then subsampled for dissolved
trace metal samples, filtrations, Fast Repetition Rate fluorometry (FRRf), and flow cytometry analysis. Samples for
nutrients and total alkalinity (TA) were transferred using the same pump but through a silicone tube into 80 mL HDPE
bottles. Total alkalinity and macronutrient samples were filtered during this process through a 0.2 µm nylon filter attached
to the silicone tube to remove all particles and organisms > 0.2 µm.

**2.4 Salinity, nutrients, carbonate chemistry, and trace metal analysis**
Salinity was measured before and at the end of the experiment using a HACH HQ40d portable meter. The $pH_T$ (total scale)
and temperatures were measured daily (2-3 hours after the onset of the light period) using a pH meter (914
pH/Conductometer Metrohm). We recorded voltages and temperature from the pH meter and calibrated the $pH_T$ at original
temperature at sampled time using the certified reference material (CRM) Tris buffer following the method described in
SOP6a by Dickson et al. (2007). Briefly, the standard buffer's pH and voltage at different temperature gradients were
recorded, and temperature vs. voltage polynomial regression data were generated for calculating calibrated pH values ($pH_T$)
(refer to Eq. 3 in SOP6a of Dickson et al. (2007)). The regression could then be used to obtain a CRM pH value for each
temperature and to calibrate the pH measured in the microcosms to the total pH scale.

Total alkalinity was sampled every four days. It was measured in duplicate using a Metrohm 862 Compact Titrosampler
coupled with an Aquatrode Plus with PT1000 temperature sensor following the SOP3b open-cell titration protocol
described in Dickson et al. (2007). Filtered TA samples were stored at 8 ℃ for a maximum of 23 days before measurement.



Titration curves were evaluated using the "calkulate" script within PyCO2sys by Humphreys et al. (2022). The carbon
chemistry equilibrium was calculated with the R package "seacarb" Gattuso et al. (2023) from $pH_T$, TA, phosphate, silicate,
temperature, and salinities using stoichiometric equilibrium constants from Lueker et al. (2000). Dissolved macronutrients
were measured every second day using standard spectrophotometric methods developed by Hansen and Koroleff (1999)
on the day the samples were taken from the microcosms.

Dissolved trace metal concentrations were measured four times during the experiment: a few hours before olivine and slag
were added, a few hours after these minerals were added on day 2, near the middle of the experiment on day 13, and at the
end of the experiment on day 22. Sixty mL of seawater was collected using an acid-washed 60 mL syringe, and the seawater
was filtered through 25 mm diameter 0.2 μm pore size polycarbonate filters. Unfortunately, we did not notice that 0.2 μm
pore size nylon filters (acid washed) were used during sampling on days 1 and 2 so we refiltered these seawater samples
again using 0.2 μm pore size polycarbonate filters after one month. All seawater samples were diluted approximately 20-
fold by weight using Milli-Q water (18.2 MΩ·cm grade) and acidified using 1 % ultrapure HCl. These samples were
analysed using Sector Field Inductively Coupled Plasma Mass Spectrometry (SF-ICP-MS) employing multiple resolution
settings to overcome major spectral interferences. Due to the presence of abundant major metal ions in our samples, such
as Na and Mg, natural open-ocean seawater from the Southern Ocean with very low trace metal concentrations was diluted
20 times with Milli-Q water water and used as a representative blank. The same Southern Ocean seawater was enriched
with different gradients of trace metal standards to calculate the samples' trace metal concentrations. Seven of the total 36
samples had abnormal trace metal or phosphate concentrations, and 4 of them were from day 1. We considered them as
contaminated using the interquartile range (IQR) criterion and excluded them from the data analysis.

**2.5 Particulate matter and plankton community analysis**
Chlorophyll *a* was sampled every second day by filtering the seawater through glass fibre filters (GF/F, pore size = 0.7 μm,
diameter =25 mm), and filters were stored in 15 mL polypropylene tubes wrapped with aluminium foil and stored at -80 °C
for 50-70 days before measurement. Each filter was immersed in 10 mL 100 % methanol for 18-20 h to extract chlorophyll
from phytoplankton and these samples were analysed on a Turner fluorometer (Model 10-AU) following the method
described by Evans et al. (1987).

Phytoplankton flow cytometry samples were fixed with 40 μL of a mixture of formaldehyde-hexamine (18 %:10 % v/w)
added to 1400 μL of seawater sample. All bacteria samples (700 μL) were fixed with 14 μL glutaraldehyde (Electron-
microscope grade, 25 %). After mixing samples with fixatives, samples were stored for 25 minutes at 10 ℃, then flash-
frozen in liquid nitrogen, and stored at -80 °C until measurement 83-86 days later. Directly before the measurement,
samples were thawed at 37 ℃. Bacteria samples were stained with SYBR green I (diluted in dimethylsulfoxide) at a final
ratio of 1:10000 (SYBR Green I: sample).

A Cytek Aurora flow cytometer (Cytek Biosciences) was used to quantify the abundance of fluorescing particles such as
phytoplankton or stained bacteria. Phytoplankton groups were distinguished based on their fluorescence signal intensity of
different laser excitation/emission wavelength combinations and forward scatter (FSC). The yellow-green laser (centre
wavelength: 577 nm), in combination with FSC signal strength, was used to separate cyanobacteria and cryptophytes from



other phytoplankton. The violet laser (centre wavelength: 664 nm) in combination with FSC was used to distinguish
picoeukaryotes, nanoeukaryotes, and microphytoplankton. The blue laser (centre wavelength: 508 nm) in combination with
FSC was used to distinguish bacteria from other living (i.e., DNA-containing) particles (Fig. S. 1).

The biovolume of each classified flow cytometry phytoplankton type was calculated using the equation:

$Biovolume = Cell\ number\ count \times (\frac{FSC}{10248})^{2.14}$                                             (4)

Where biovolume is the biovolume of the phytoplankton ($\mu m^3$), cell number is the cell count per mL of sample, and the
FSC is the forward scatter signal value from the flow cytometry. This equation is calculated based on the relationship
between biovolume and FSC for different phytoplankton species (Selfe, 2022).

Phytoplankton photosynthetic performance was estimated from the rapid light curves measured with an FRRf (FastOcean
Sensor FRRf3, Chelsea Instruments Group) every second day following the protocol adapted from Schallenberg et al.
(2020). Samples were kept in the dark for 20 minutes before the measurement and then added to the FRR fluorometry
cuvette, which was temperature-controlled at 13.5 °C. Filtered natural seawater was used for blank correction. The channel
with different light wavelengths (450, 530, and 624 nm) was used in each acquisition sequence. At least 10 acquisitions
were measured for each sample. The maximum electron transport rate ($ETR_{max}$), initial slope of the rapid light curve ($\alpha$),
and the light-saturation parameter ($E_k$) were calculated using the equation described by Platt et al. (1980) without
photoinhibition:

$ETR = ETR_{max}\left[1 - e^{-\frac{\alpha E}{ETR_{max}}}\right]$                                                 (5)

These parameters together with the maximum quantum yield of PSII ($F_v/F_m$) were used to compare the photosynthetic
performance of the phytoplankton communities in different microcosms.

Seawater was sampled before the treatment and at the end of the experiment for particulate trace metal concentrations.
Samples of 100 mL were filtered through an acid-cleaned polycarbonate filter (25 mm diameter, 0.8 μm pore size) and
placed in an acid-cleaned polypropylene filter holder on a trace metal-clean bench. The filters were washed with the EDTA-
oxalate reagent (1.4 mL) twice (8 min total) and rinsed with chelexed NaCl solution (0.6 mol L$^{-1}$ with 2.38 mmol L$^{-1}$ of
HCO$_3^-$, pH=8.2) 10 times (1.5 mL aliquots) (Tang and Morel, 2006). Filters were stored in acid-washed well plates at -
20 °C before analysis. The digestion process followed the method reported by Bowie et al. (2010). Briefly, all samples and
triplicate certified reference materials plankton standards (50 mg/vial) were digested in a mixture of strong ultrapure acids
(750 μL 12 mol L$^{-1}$ HCl, 250 μL 40 % HF, 250 μL 14 mol L$^{-1}$ HNO$_3$) in 15 mL Teflon perfluoroalkoxy (PFA) vials on a
95 °C hot plate for 12 h in a fume hood. They were then dry evaporated for 4 h and re-suspended in 10 % v-v ultrapure
HNO$_3$. All prepared solutions had indium as internal standard added to a final concentration of 10 μg L$^{-1}$. Three pre-mixed
multi-element standard solutions (MISA) were prepared as external calibration standards.

Particulate organic carbon (POC) was sampled by filtering 100 mL of seawater from each microcosm. Glass fibre filters
(Whatman GF/F, pore size =0.7 μm, diameter =13 mm) were pre-combusted at 400 °C for 6 h. Filters were stored at -20 °C



before measurement. Samples were treated via fuming with 2N HCl to remove carbonates overnight and dried in the oven
for 4h. Finally, filters were folded into silver cups and stored in a desiccator until analysis. Samples were analysed for
carbon with a Thermo Finnigan EA 1112 Series Flash Elemental Analyser (CSL, University of Tasmania).

Biogenic silica (BSi) concentrations were analysed every 4 days by filtering 100 mL of seawater from each microcosm.
Mixed Cellulose Ester (MCE) membrane filters (diameter = 25 mm, pore size = 0.8 µm) were used for BSi samples. BSi
filters were placed in a plastic petri dish and stored at -20 ℃ before measurement. Filters were processed using the hot
NaOH digestion method of Nelson et al. (1989). The final solution was measured using the same process as the dissolved
silicate (see section 2.4).

A self-made plastic zooplankton net (20 mm height and 15 mm width) with a 210 µm mesh size was acid-washed first and
then used to collect zooplankton from microcosms before mineral addition on day 2, near the middle (day 13), and at the
end of the experiment (day 23). Samples were stored in 10 % formalin seawater solutions and kept at room temperature
until measurements. Zooplankton were quantified and identified under a Leica M165C microscope fitted with a Canon 5D
camera. The number of zooplankton from one mini-trawl in each collection was converted to the unit of individual $L^{-1}$ and
used for data analysis. The diversity of zooplankton communities was estimated with the Shannon Diversity Index (H)
calculated as:

$$H = -\sum(pi \times \ln(pi)) \tag{6}$$

where pi is the proportion of the entire zooplankton community made up of individual species abundance, and ln is the
natural logarithm.


**2.6 Statistic analysis**
R studio was used for data analyses. Generalized additive models (GAMs) from the package "mgcv" were fitted to the data
to predict the changes over time. The GAMs all shared the same equations:

$$Y = s(Day), \tag{7}$$

in which Y presents the dependent variable and s(Day) is the smooth term of the day of the experiment. Another GAM was
used to detect significant differences between treatments and the control:

$$Y = s(Day) + s(Day, by = oTreatment) \tag{8}$$

In this equation, the variable "Treatment" includes three conditions: "Control", "Slag" and "Olivine"; while "oTreatment"
is the ordered factor of the variable "Treatment" which allowed us to compare the GAMs smooth terms from different
treatments and the control (Simpson, 2017).



For the analysis of trace metal concentrations and zooplankton abundance, Generalized Linear Models (GLMs) from the
'stats' package were fitted to the data to determine significant differences between treatments and the control. The selection
of specific GLMs was based on the distribution of the raw data. One GLM equation is

$$Y = Treatment + \frac{Day}{22} + (\frac{Day}{22})^2 \qquad (9)$$

with family = Gamma, where Y represents the measured parameter (abundance of a zooplankton species and dissolved
trace metal concentrations); treatment is the conditions ("Control", "Slag" and "Olivine"); and Day represents the day of
the experiment. The other GLM equation,

$$Y = Treatment + Day \qquad (10)$$

with family = Gaussian, was employed for particulate trace metal data and the Shannon Diversity Index. To compare the
contribution of the three treatments on the measured parameters, Tukey's significant difference test was conducted on the
GLMs using the 'glht' function.

**3. Results**
**3.1 Elemental composition and grain size of the finely-ground minerals**
SEM analysis revealed the approximate elemental composition of olivine and slag powder (Table 1). Based on this analysis
the olivine composition resembles the Mg-rich olivine mineral "forsterite" ($Mg_2SiO_4$). The particle size spectrum of olivine
powder is shown in detail in Fig. S2. Roughly 69 % of the olivine particles, when measured by volume, fell within the
diameter range of 35 - 300 μm. Additionally, SEM analysis revealed high levels of Ca and O in the slag, indicative of the
considerable $Ca(OH)_2$ and CaO content of the powder (Table 1; please note that H cannot be measured with the applied
method). The particle size measurement (Fig. S2) showed that 78 % of the ground slag particles were between 35 - 300
μm.

**Table 1.** The weight percentage of elements from two minerals.  Unit: wt %.

| Element | O | Ca | Mn | Si | Mg | Fe | Al | Ti | Cr | Ni |
|---------|------|------|-----|------|------|------|-----|-----|-----|-----|
| Olivine | 39.9 | 0.4 | | 19.9 | 26.4 | 13.0 | 1.0 | | | 0.8 |
| Steel slag | 41.9 | 36.0 | 7.0 | 6.5 | 4.3 | 3.7 | 3.4 | 1.7 | 1.6 | |



**3.2  Physical and chemical conditions over the course of the experiment.**
On day 2 of the experiment, when olivine particles were introduced into the microcosms, the smallest fraction of the powder
remained suspended, causing the seawater to become highly turbid for several days. The resulting milky appearance of the
seawater eventually faded over a period of approximately five days, and by day 5, the turbidity had visually become like



the slag treatment and the control. This effect was not anticipated, and as a result, we decided to investigate its impact on
light intensity. To do so, a test was conducted after the main experiment in which olivine powder was added to a microcosm
identical to those used in the experiment, and light intensity was measured daily at a depth of 0.15 m. The results showed
that the addition of olivine caused an initial reduction in light intensity of 18.5 % at 15 mins after addition, which declined
to 7.4 %, 3.7 %, 3.7 % and 0 % after 1, 2, 3, and 4 days, respectively. These findings indicate that olivine additions can
significantly affect the light environment in the microcosms, whereas no such effect was observed in the slag treatment.

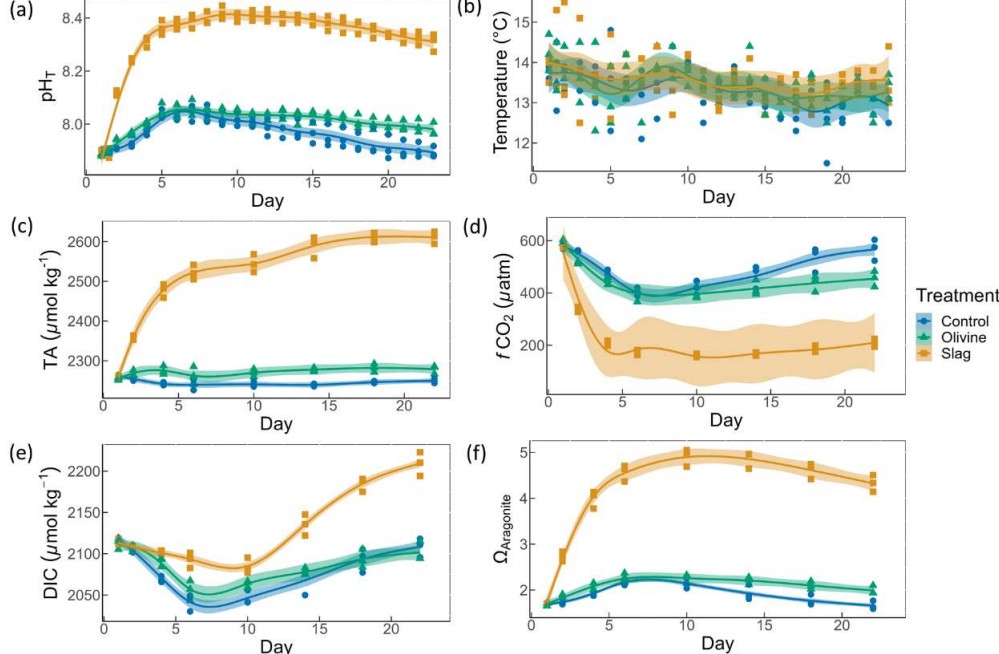


**Fig. 2**. Carbonate chemistry conditions. The temporal development of (a) $pH_T$, (b) temperature, (c) total alkalinity (TA), (d) $CO_2$ fugacity
($fCO_2$) computed at *in situ* temperature and atmospheric pressure, (e) dissolved inorganic carbon (DIC), and (f) aragonite saturation state
($\Omega_{aragonite}$). The dots represent the raw data (n=3 for each treatment per sampling time), and the fitted curve is the generalized additive
model (GAM). The shading represents the 95 % confidence interval of the fitted GAM.

The $pH_T$ of all microcosms increased from day 1 to day 5 (Fig. 2a). This was partially due to photosynthetic $CO_2$ drawdown
during phytoplankton blooms that commenced in each microcosm. During the peak of the bloom, olivine addition led to a
slightly higher $pH_T$ (8.054 ± 0.014, average values ± standard error) than the control (8.037 ± 0.010), but $pH_T$ remained at
considerably higher levels in the olivine treatment after the bloom compared to the control. The slag addition increased the
$pH_T$ in the microcosm from initially 7.897 (± 0.001) to 8.411 (± 0.015), which was significantly higher than the olivine
treatment and the control throughout the experiment. The final $pH_T$ of the control, olivine, and slag treatments were 7.893
± 0.012, 7.978 ± 0.015, and 8.309 ± 0.019, respectively.

When comparing GAMs, P-means represent the p-value obtained from comparing two GAMs, such as the control and the
olivine treatment. If P-means is below 0.05, it indicates that the mean values of the two GAMs exhibit significant



differences over the course of the experiment. Conversely, if P-means is equal to or greater than 0.05, it suggests that the
two GAMs have similar mean values. In contrast, P-smooths represents the p-value derived from comparing the smooth
terms of two GAMs. If P-smooths is below 0.05, it indicates that the two GAMs demonstrate significantly different trends
in their change over time. In our analysis, all the fitted GAMs from the treatments and the control exhibited significant
differences in $pH_T$ from each other, as evidenced by the p-values of both P-means and P-smooths being smaller than 0.001.
For detailed results of the GAM p-values, please refer to Table S1.

Total alkalinity increased marginally from $2255 \pm 2$ to $2262 \pm 13$ µmol kg$^{-1}$ within the first 6 days after olivine addition
while it increased more substantially from $2259 \pm 1$ to $2522 \pm 11$ µmol kg$^{-1}$ in the same time span in the slag treatment (Fig.
2c). The TA in the control decreased from $2261 \pm 2$ µmol kg$^{-1}$ to $2240 \pm 7$ µmol kg$^{-1}$ from day 1 to day 6 but remained
stable thereafter. The TA reached $2279 \pm 6$ µmol kg$^{-1}$ in the olivine treatment and $2611 \pm 9$ µmol kg$^{-1}$ in the slag treatment
group on day 22. The slag treatment reached a significantly higher TA than the olivine treatment and the control (P-
smooths<0.001). The mean TA from GAM in olivine treatment was higher than the control group (P-means<0.001).

The $CO_2$ fugacity ($fCO_2$) computed at *in situ* temperature and atmospheric pressure decreased continuously in the first 6
days in all microcosms (Fig. 2d). Then it increased again in the control and olivine treatments while staying lower in the
slag treatment. Dissolved inorganic carbon (Fig. 2e) and the aragonite saturation state ($\Omega_{aragonite}$; Fig. 2f) revealed a similar
trend over the course of the experiment in the control and the olivine treatment. In contrast, the slag treatment had higher
DIC and $\Omega_{aragonite}$ values throughout the experiment.


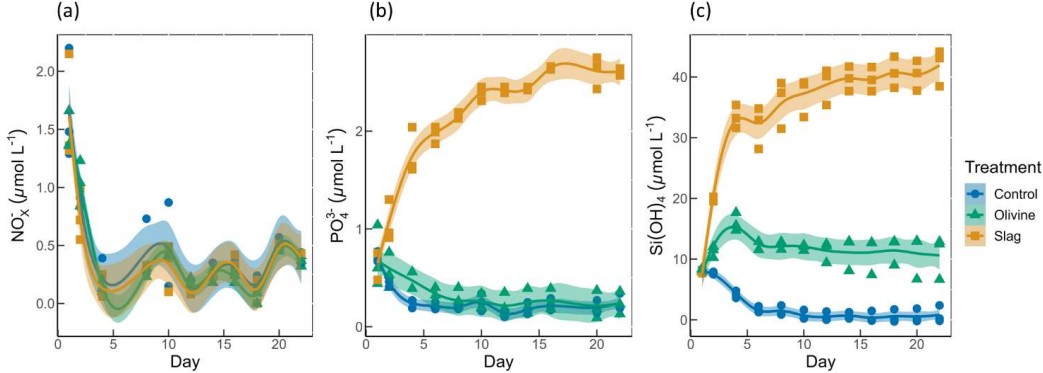


**Fig. 3.** Macronutrients concentrations over the course of the study. (a) Nitrate and nitrite concentrations. (b) Phosphate concentrations.
(c) Silicic acid concentrations. The dots represent the raw data (n=3 for each treatment per collection), and the fitted curve is the
generalized additive model.

Initial nitrate and nitrite ($NO_x^-$), phosphate ($PO_4^{3-}$), and silicic acid ($Si(OH)_4$) concentrations were $1.58 \pm 0.12$, $0.69 \pm 0.59$,
and $8.04 \pm 0.10$ µmol L$^{-1}$, respectively (Fig. 3). $NO_x^-$ declined rapidly in all microcosms once the experiment had
commenced to values below 0.5 µmol L$^{-1}$ and no significant difference was detected between treatments and control (P-
smooths >0.05; Fig. 3a). In both the olivine treatment and the control, the $PO_4^{3-}$ concentration decreased in the first six





days (Fig. 3b). In the slag treatment, $PO_4^{3-}$ increased to a maximum of $2.65 \pm 0.01$ µmol $L^{-1}$, which was significantly higher
than in the olivine treatment and the control (P-means <0.001). The $Si(OH)_4$ concentration increased to a maximum of
$15.99 \pm 0.87$ µmol $L^{-1}$ in the olivine treatment, increased to a maximum of $41.92 \pm 1.75$ µmol $L^{-1}$ in the slag treatment, but
decreased below the detection limit in the control (Fig. 3c). Significant differences were observed in the development of
$Si(OH)_4$ between all treatments and the control (Table S1).

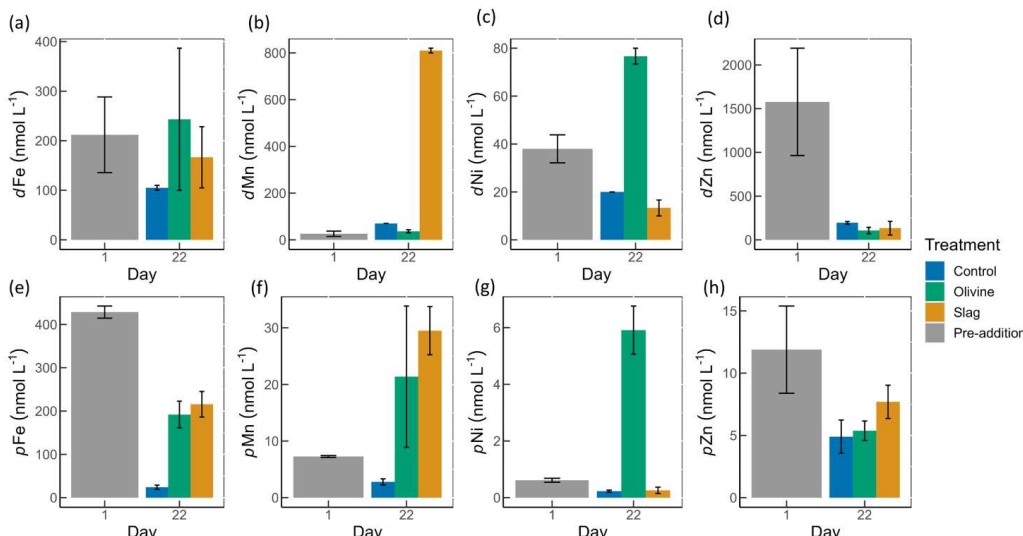


**Fig. 4.** Dissolved and particulate trace metal concentrations in microcosm seawater. (a)-(d) are dissolved trace metal concentrations, and
(e)-(h) are total particulate trace metal concentrations. The error bars represent the standard error from measured samples. The pre-
addition data shown in (a)-(d) represent the average of 5 microcosms before addition of slag or olivine. The data for the control on day
22 in (a)-(d) and for the pre-addition on day 1 in (e)-(h) were based on two of three microcosm replicates. The remaining data were based
on all three microcosm replicates.

After 21 days of experiment, the treatments showed a significant increase in dissolved Al concentrations from $504 \pm 80$ to
$970 \pm 228$ nmol $L^{-1}$ in olivine treatment, and from $504 \pm 80$ to $1093 \pm 77$ nmol $L^{-1}$ in slag treatment, while in the control
dissolved Al decreased to $230 \pm 10$ nmol $L^{-1}$ (Fig. S3). The fitted GLMs were compared, and the p-value revealed how
much influence a treatment had on the dissolved metal concentrations (Table S2). The results indicate that the slag and
olivine additions led to significantly higher Al concentrations than in the control (p-values < 0.05), but no significant
difference was found between the two treatments (p-value = 0.189). The olivine treatment released Cu into the seawater,
and the Cu concentration in the olivine on day 22 was significantly higher than the slag treatment and the control (p-value
<0.05) (Fig. S3).The addition of olivine and slag released some Fe, but overall, the concentration of Fe did not differ
between treatments (Fig. 4a). The slag released a substantial amount of dissolved Mn (maximum $820 \pm 10$ nmol $L^{-1}$ on
day 22) (Fig. 4b), leading to significantly higher concentrations than in the olivine treatment and the control (p-values <
0.001). A significant amount of dissolved Ni was released from the olivine powder (p-values <0.001) (Fig. 4c). The initial
concentration of dissolved Zn in seawater was much higher than on day 22 in all microcosms, and no significant difference



in Zn concentrations was found between the treatments and the control (Fig. S3).

Particulate concentrations of some trace metals also differed between treatments. The total particulate Fe decreased in all
microcosms on day 22 comparing with the pre-addition level, but both mineral addition treatments had higher particulate
Fe concentrations than the control on day 22 (Fig. 4e). The addition of slag elevated particulate Mn concentrations to a
level higher than the pre-addition and the control on day 22 (Fig. 4f), while the addition of olivine increased the particulate
Ni concentrations to a level higher than the slag, the control, and the pre-addition (Fig. 4g). The particulate Zn
concentrations in general decreased by the end of the experiment (Fig. 4h), and no significant differences were found
between the treatments and the control (Table S2).

The POC on day 1 and day 22 from all microcosms were very similar, $10.99 \pm 0.58$ and $11.03 \pm 0.41$ µmol L$^{-1}$ respectively
(Fig. S4) so the metal:POC results were consistent with the particulate trace metal results (Fig. 4 e-h). In general, the non-
surface metal:POC are positively correlated with the total metal:POC ratios (Fig. S5). The ratio of non-surface to total
particulate trace metal concentrations is summarized in Table S4. Both non-surface and total Fe concentrations decreased
in microcosms on day 22 compared with the pre-addition level. Iron:POC ratios were significantly higher in the treatments
than in the control on day 22 (p-values $<0.05$. Table S2), and there was no significant difference between mineral addition
treatments. The non-surface to total Fe:POC ratios were $> 0.94$ in all microcosms on both day 1 and day 22. The total and
non-surface Mn:POC ratio was the highest in the slag treatment. These ratios were higher than the pre-addition level and
the control at the end of the experiment. The total particulate Ni concentrations in the olivine treatment were significantly
higher than before olivine addition. The olivine treatment led to a $>22$-fold higher Ni:POC ratio compared to the other two
treatments (p-value $<0.001$).

## 3.3 Development and physiology of the plankton community

**Fig. 5.** Temporal development of chlorophyll-a concentration (chl-a), BSi, and different eukaryotic and bacterial plankton groups as determined with flow cytometry. (a) chlorophyll-a; (b) BSi; cell concentrations of (c) heterotrophic bacteria, (d) microphytoplankton, (e) nanoeukaryotes2, (f) nanoeukaryotes1 (g) picoeukaryotes, (h) cyanobacteria, and (i) cryptophytes; biovolume proportion of (j) microphytoplankton, (k) nanoeukaryotes2, (l) nanoeukaryotes1 (m) picoeukaryotes, (n) cyanobacteria, and (o) cryptophytes. The fiigure data points represent the raw data, and the fitted curve is the generalized additive model. The shaded area represents the 95 % confidence interval.

The chl-a concentration in all microcosms increased from day 1 to day 4 from 1 µg L⁻¹ to 3-4 µg L⁻¹ (Fig. 5a). The chl-a



429 concentration then decreased rapidly from day 4 to day 8, then continued to decrease, though more slowly, to <0.3 µg L⁻¹

430 until the end of the experiment. The GAMs of chl-a did not show any difference between treatments and the control (both

431 P-means and P-smooths >0.05, see Table S1).

432

433 The BSi concentration increased from day 1 to day 6 in all microcosms (Fig. 5b). In the olivine and slag treatments, BSi

434 concentrations decreased slightly after the peak until day 12 but then increased again. In contrast, BSi concentration

435 decreased continuously in the control after the initial peak. Olivine particles suspended in seawater after the mineral

436 addition (see section 3.2) partially ended up on BSi filters during filtration. This led to extremely high BSi measurements

437 on days 2 and 4 that were removed from Fig. 5b. Without these outliers, the mean of fitted BSi GAM in the olivine treatment

438 was lower than the control and the slag treatment (Table S1), and the slag had the highest average BSi over the course of

439 the experiment. Overall, the BSi trends in the two treatments were similar (P-smooths = 0.269), although both were

440 significantly different from the control (P-smooths <0.05).

441

442 The development of the phytoplankton community composition showed significant differences between the treatments and

443 the control. In general, most phytoplankton groups exhibited similar patterns to chl-a, with peak cell numbers occurring on

444 day 4 (Fig. 5f-i). Following day 6, phytoplankton cell abundance generally decreased steadily. Microphytoplankton

445 displayed similar trends to the results for BSi. Before day 10, all microcosms had similar microphytoplankton abundances

446 (Fig. 5d). However, in the control group, microphytoplankton abundance declined continuously and at a faster rate

447 compared to the other two treatments (P-smooths values <0.03). From day 2 to day 6, the abundance of nanoeukaryotes1,

448 nanoeukaryotes2, picoeukaryotes, and cryophytes was higher in the olivine treatment compared to the slag treatment and

449 the control. After day 8, their abundance decreased to a similar level as the other two groups. Notably, there were few

450 significant differences observed between the slag treatment and the control group in terms of the abundances of

451 nanoeukaryotes1, nanoeukaryotes2, picoeukaryotes, and cryptophytes throughout the experiment. In the olivine treatment,

452 cyanobacteria experienced a second bloom after day 10, which was significantly different from the other two groups (P-

453 smooths <0.01). Heterotrophic bacteria exhibited an increase and decline pattern following the phytoplankton bloom until

454 day 8 (Fig. 5c). Subsequently, bacteria abundance increased again, reaching a second peak during days 12-14, followed by

455 a decline until the end of the experiment. The decline in bacteria abundance was slower in the olivine treatment, although

456 no significant differences were detected between treatments (Table S1).

457

458 Among all the microcosms, microphytoplankton, nanoeukaryotes2, and cryptophytes consistently accounted for the largest

459 proportion of biovolume. From the perspective of biovolume proportion, the mineral addition mainly influenced the

460 microphytoplankton and nanoeukaryotes. The control had similar phytoplankton biovolume distribution as the treatments

461 from day 1 to day 15, but after that the proportion of microphytoplankton biovolume decreased to a level significantly

462 lower than the treatments. In the control treatment, the proportion of nanoeukaryotes' biovolume increased as the proportion

463 of microphytoplankton decreased. The biovolume of picoeukaryotes, cyanobacteria and cryptophytes increased during the

464 phytoplankton bloom and then decreased drastically after the bloom. There were no significant differences in biovolume

465 observed for picoeukaryotes, cyanobacteria and cryptophytes between the treatments and the control.

466



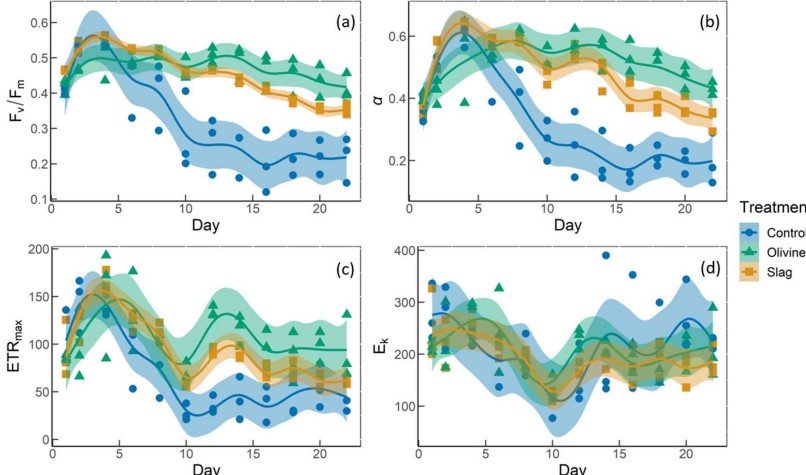

**Fig. 6.** The photosynthetic performance of the phytoplankton community. (a) $F_v/F_m$, the maximum quantum yield of photosynthesis II. (b) $\alpha$, the initial slope of the rapid light curves. (c) $ETR_{max}$ is the maximum electron transport rate, the maximum potential photosynthetic rate. (d) $E_k$ is light-saturation parameter, Unit: $\mu$mol photons $m^{-2}$ $s^{-1}$.

The temporal development of $F_v/F_m$, $\alpha$, $ETR_{max}$, and $E_k$ is illustrated in Fig. 6. The $F_v/F_m$ values of the phytoplankton community were approximately $0.42 \pm 0.01$ and increased to levels > 0.5 during the peak of the phytoplankton bloom on day 4 (Fig. 6a). Following the bloom, $F_v/F_m$ values dropped below 0.3 in the control. However, the decline in $F_v/F_m$ after the bloom was less pronounced in the two mineral addition treatments. At the end of the experiment, $F_v/F_m$ was $0.22 \pm 0.04$ in the control, $0.35 \pm 0.01$ in the slag treatment, and $0.42 \pm 0.02$ in the olivine treatment. The temporal development of $\alpha$ aligned with the patterns observed for $F_v/F_m$ (compare Fig. 6a and 6b). The maximum values of $ETR_{max}$ were observed on day 4 in the control and the slag treatment, while in the olivine treatment, it occurred on day 5 (Fig. 6c). Subsequently, $ETR_{max}$ continuously decreased until day 10 and then stabilized until the end of the experiment. However, $ETR_{max}$ exhibited a subsequent increase in the mineral treatments around day 12. The $ETR_{max}$ values were higher in the mineral treatments compared to the control group (P-means <0.001, Table S1). The parameter $E_k$ decreased from $246 \pm 17$ $\mu$mol photons $m^{-2}$ $s^{-1}$ on day 1 to $121 \pm 7$ $\mu$mol photons $m^{-2}$ $s^{-1}$ on day 10, and then it increased again to approximately 200 $\mu$mol photons $m^{-2}$ $s^{-1}$ by the end of the experiment (Fig. 6d). The change in $E_k$ did not exhibit significant differences between the treatments and the control (both P-means and P-smooths >0.05).





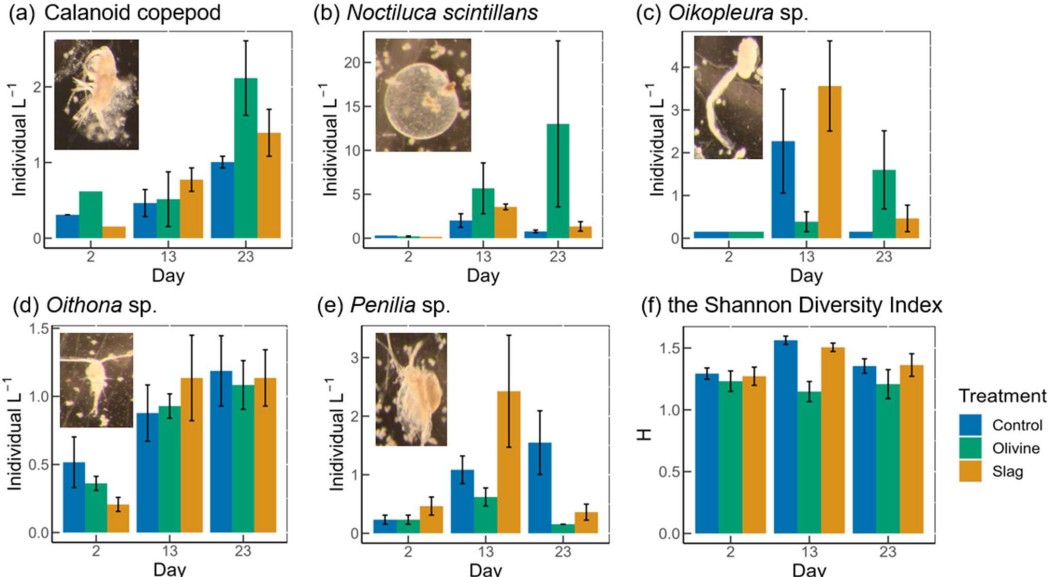

**Fig. 7.** The dominant zooplankton abundance and community diversity from different treatments. Abundance of dominant zooplankton in microcosms: (a) calanoid copepod; (b) *Noctiluca scintillans*; (c) *Oikopleura* sp.; (d) *Oithona* sp.; (e) *Penilia* sp.; and (f) the Shannon diversity index (H) of different treatments and the control. Error bars represent the standard error calculated from three microcosm replicates. Photographs of each zooplankton group are shown on the corresponding graphs.

Thirteen zooplankton taxonomic groups were identified in the microcosms. The dominant taxa were the appendicularian *Oikopleura* sp., the cyclopoid copepod *Oithona* sp., the cladoceran *Penilia* sp., the heterotrophic dinoflagellate *N. scintillans* and several calanoid copepods including *Acartia* sp., *Paracalanus* sp. and *Gladioferens* sp. The larvae and eggs of *Oikopleura*, *Penilia* and copepod were also observed under the microscope. In general, higher zooplankton numbers were observed after the bloom on day 13 (Fig. 7). The abundance of calanoid copepods and *Oithona* sp. increased after day 2 (Fig. 7 a, d), and there was no significant difference between treatments and the control (p-values >0.05, Table S3). The abundance of *N. scintillans* increased significantly more in the olivine treatment than in the control and the slag treatment, with highest abundance of $13 \pm 9$ individual $L^{-1}$ observed in the olivine treatment on the last day (Fig. 7 b). The abundance of *Oikopleura* in the control and the slag treatment was higher than the olivine treatment on day 13 but was higher in the olivine treatment on day 22 (Fig. 7c). A higher abundance of *Penilia* sp. was found in the slag treatment on day 13 and in the control on day 23 (Fig. 7e). Due to the patchy distribution of zooplankton, these data have large standard errors and only the differences in the numbers of *N. scintillans* in the olivine treatment were statistically significantly different from the slag treatment and the control (p-value <0.05, Table S3).

Considering the control and slag treatment, the Shannon Diversity Index (H) increased from day 2 to day 13 and declined on day 23, while in the olivine treatment, H was lower on day 13 than on day 2 and day 23 (Fig. 7f). The GLMs revealed that the olivine treatment had significantly lower H on day 13 than the control and the slag treatment (p-values <0.001). There were no significant differences in H between the control and the slag treatment (Table S3). The addition of olivine decreased the zooplankton community's diversity. This is mainly driven by distinct trends observed in the abundance of *Oikopleura* sp., *Penilia* sp., and *N. scintillans* (Fig. 7).




## 4. Discussion

### 4.1  $CO_2$ removal potential of slag and olivine

The slag powder created significantly higher $CO_2$ removal potential than the olivine powder over the course of the study.
$Ca(OH)_2$ and $CaO$ in slag and $Mg_2SiO_4$ in olivine are likely to be the main functional minerals driving the measured
alkalinity enhancement. Total alkalinity increased by 361 µmol kg$^{-1}$ in the slag treatment while it increased by only 29
µmol kg$^{-1}$ in the olivine treatment, equivalent to a potential increase of the marine inorganic carbon by 14.7 and 0.9 %
within 3 weeks of their application. When normalizing these alkalinity increases to the same material weight, 1 g of slag
would release 9626 µmol TA while 1 g of olivine would release 16 µmol TA. Thus, over 3 weeks of experimental incubation,
slag is ~600-fold more efficient in releasing alkalinity for particles of this size class (please note that particle size spectra
of olivine and slag were similar but not identical; Fig. S1). We can also use these values to make a rough estimate of how
much $CO_2$ these two minerals could potentially sequester. One mole of alkalinity from olivine and slag can sequester
approximately 0.85 mole of $CO_2$. Thus, one tonne of slag and olivine powder as used here could sequester 360 and 0.6 kg,
respectively, within 3 weeks. It is likely that optimization of particle size and application method may lead to higher
efficiencies. Nevertheless, the slag showed potential as an OAE source mineral, even when applied as relatively coarse
powder in this experiment.

### 4.2 OAE impacts on phytoplankton physiology and community

The chlorophyll-a concentration was indistinguishable between treatments and the control suggesting a limited effect of
slag- or olivine-based OAE on phytoplankton bloom dynamics under these experimental settings. The phytoplankton
community was most likely N-limited after day 4 so that the release of $Si(OH)_4$ from olivine and $Si(OH)_4$ and $PO_4^{3-}$ from
slag did not stimulate a further increase in chlorophyll-a concentration in the treatments. The development of BSi
concentrations is indicative of the prevalence of diatoms in the microcosms but differences between treatments and the
control were small. The release of $Si(OH)_4$ through olivine and slag will most likely benefit diatoms but this fertilization
effect did not manifest in this specific experiment because N was limiting diatom growth. However, when new N is supplied
then diatoms will likely take a bigger share of the limiting N pool when olivine or slag are used for OAE. As such, diatoms
are likely to benefit from olivine and slag applications, as has been shown in $Si(OH)_4$ manipulation experiments outside
the context of OAE research (Egge and Jacobsen, 1997). In the case of slag, the release of $PO_4^{3-}$ will likely be another
driver that affects plankton productivity and community composition. As for $Si(OH)_4$, however, the effect of additional
$PO_4^{3-}$ did likely not materialise in this experiment because $PO_4^{3-}$ was not limiting over the course of the study. However, in
ecosystems where $PO_4^{3-}$ is a limiting resource, the application of slag could enhance productivity with associated benefits
for higher trophic levels. In contrast, excessive applications of slag and concomitant $PO_4^{3-}$ release could also pose a risk of
eutrophication. Future studies may need to investigate what the most sustainable dose of OAE via olivine and/or slag
applications could be.

The flow cytometry results further revealed the change in phytoplankton community composition. Both the olivine and



slag treatments sustained higher microphytoplankton abundances after the peak of the phytoplankton bloom on day 6. This
trend is consistent with some photophysiological parameters such as $F_v/F_m$ so that it is tempting to assume that
photophysiological fitness gain measured with the FRRf led to higher competitiveness of microphytoplankton in the
community. Indeed, calculations of biovolume with flow cytometry data indicate that microphytoplankton were
predominantly contributing to the phytoplankton community biovolume so that the responses measured by the FRRf were
to a large extent driven by this group.

Apart from the increased microphytoplankton abundance, for the slag treatment, other phytoplankton groups distinguished
with flow cytometry did not deviate considerably from the control. The olivine addition, however, triggered more
pronounced shifts in the phytoplankton community. In particular, the nanoeukaryotes (roughly between 2-20 μm),
picoeukaryotes and the cryptophytes showed relatively higher abundance during the peak of the phytoplankton bloom, and
the abundance of cyanobacteria was higher after the bloom. We speculate that this shift following olivine treatment may
be attributable to a top-down effect from the decrease in zooplankton grazing effects in microcosms, which will be
discussed in section 4.3.

The measurement of photophysiological parameters revealed that the phytoplankton had generally better photosynthetic
performance in the slag and olivine treatments than in the control, especially after the phytoplankton bloom. During the
first 5 days, the changes in phytoplankton photosynthetic performance was indistinguishable amongst treatments. All
microcosms had similar health because of the relatively high $NO_x^-$ concentrations and Fe supply (around 100 nmol L$^{-1}$).
After day 5, the $F_v/F_m$, α and $ETR_{max}$ values decreased significantly faster in the control than in the treatments, and to values
lower than the initial condition. A decrease of $F_v/F_m$ is commonly associated with physiological stress, such as nutrient
limitation, and high light stress (Bhagooli, et al. 2021), with Fe limitation causing a more pronounced decline in $F_v/F_m$ than
nitrogen limitation (Gorbunov, et al 2021). $ETR_{max}$, which represents the maximum electron transport rate, has also been
shown to be negatively affected when phytoplankton experience nitrogen or Fe limitation (Kolber et al. 1994; Gorbunov
& Falkowski 2021). Furthermore, the change in photosynthesis performance after day 10 was suspected to be driven by
the microphytoplankton because the decrease of $F_v/F_m$, α, and $ETR_{max}$ in the control was coupled with the decrease of
microphytoplankton abundance while the other phytoplankton groups were in low abundance as in the mineral addition
treatments, and the microphytoplankton contributed significantly (75 %) to community biovolume. All microcosms were
similarly $NO_x^-$ limited from day 5 onward (Fig. 3) so that N-limitation is unlikely to explain different trends in
photophysiological parameters between the control and OAE treatments. Trace metals, especially Fe, released through slag
and olivine additions could potentially explain these differences.

Several of the trace metals released from slag and olivine are required for photosynthesis. For example, Fe is required for
many proteins functioning in photosynthesis, such as cytochromes, ferredoxin, and superoxide dismutase (SOD) (Twining
and Baines, 2013), and the addition of Fe can stimulate the growth of phytoplankton (Sunda and Huntsman, 1997) and
increase $F_v/F_m$ (Behrenfeld et al., 2006). The dissolved and particulate Fe concentrations were higher in mineral addition
treatments than in the control indicating potentially more Fe available to sustain phytoplankton photosynthesis. While this
explanation is intriguing for the observed trends in photophysiology, it remains unclear why such strong differences
occurred between mineral addition and control treatments despite dissolved Fe concentrations of ~100 nmol L$^{-1}$ at the end
of the experiment in the control. In Fe-limited ocean regions, dissolved Fe is at least two orders of magnitude lower, and



the enhancement of Fe to ~1.5 nmol L$^{-1}$ can induce major phytoplankton blooms and relieve photophysiological stress (De
Baar et al., 2005). It is possible that these coastal phytoplankton species have higher Fe requirements than those from the
open ocean where Fe is limiting (Strzepek and Harrison, 2004). We speculate that when Fe was consumed during the
phytoplankton bloom, bioavailable Fe was much lower in the control, and may have been insufficient to meet the cellular
requirements of coastal phytoplankton. Our findings therefore suggest that Fe perturbations is not only relevant for lower
Fe open ocean regions but could also be relevant for coastal ocean locations.

Alternatively, the addition of Mn, Ni and other trace metals from mineral addition may have benefited photosynthesis.
Manganese is required for the water-splitting reaction of photosystem II (Armstrong, 2008), and both Mn and Ni are
common bioactive trace metals for SODs in marine phytoplankton. The noxious superoxide anion radical ($O_2^-$) generated
from aerobic respiration and oxygenic photosynthesis could be harmful to phytoplankton physiology, and SOD removes
$O_2^-$, thus improving photosynthesis (Wafar et al., 1995; Wolfe-Simon et al., 2005). This is consistent with our
photosynthetic measurements. Interestingly, although the amounts and types of trace metals released from the slag and
olivine powders were different, they led to relatively similar $F_v/F_m$ values with only slightly higher $F_v/F_m$ in the olivine
than the slag treatment from days 10-21. Over this time, these trace metal additions could have fertilized different
phytoplankton species because different phytoplankton could have different trace metal requirements, such as for SOD.
For example, cyanobacteria have NiSOD, diatoms have MnSOD, dinoflagellates have both FeSOD and MnSOD (Wolfe-
Simon et al., 2005). Another explanation is that phytoplankton in the control were limited by bicarbonate while the
treatments had sufficient bicarbonate from added minerals. However, we were unable to determine the species-level
changes in the phytoplankton community, and hence whether these trace metals, individually or combined, could account
for the observed phytoplankton community photosynthetic performance.

**4.3 OAE impacts on the zooplankton community**
Slag-based OAE did not significantly influence the zooplankton community composition while olivine-based OAE induced
some statistically significant effects, including a lower Shannon diversity. The increase in *N. scintillans* abundance and the
decrease in *Penilia* sp. and *Oikopleura* sp. in the olivine treatment indicate that the zooplankton response to OAE can vary
among different zooplankton types.

The observed lower abundance of *Oikopleura* sp. on day 13 in the olivine treatment may indicate a temporary suppression
or a slower growth rate of this zooplankton species in response to the olivine addition. This could be attributed to the
potential effects of olivine on the availability of essential nutrients or changes in the physicochemical environment of the
water. However, the subsequent increase in *Oikopleura* sp. abundance by day 22 suggests that the growth of this species
recovered or accelerated in the olivine treatment, leading to a higher abundance compared to the slag treatment and the
control on day 22. As discussed in section 4.2., reduced *Oikopleura* sp. abundance was unlikely due to reduced food
availability since phytoplankton within the preferred edible size spectrum, such as cyanobacteria and nanoeukaryotes, were
even more abundant in the olivine treatment. Instead, we hypothesize it to be an effect of the suspended olivine particles
that occurred for approximately the first 5 days of the study that were so plentiful that they turned the enclosed seawater
milky and may have clogged the mucous feeding mesh of *Oikopleura* sp (Lombard et al., 2011).




The abundance of *Penilia* sp. was lower in the olivine treatment than the other two groups throughout the experiment while
the abundance of *N. scinitallans* was consistently higher. We cannot provide a particularly convincing hypothesis about
what specifically drove these differences though it is tempting to speculate that suspended particles present at the beginning
may have played a role also for those organisms since this was the only apparent systematic difference to the control and
slag treatment. The proliferation of *N. scintillans* can be problematic since heterotrophic dinoflagellate blooms can regulate
phytoplankton communities, cause toxicity to aquatic fish, and create an hypoxic sub-surface zone (Baliarsingh et al., 2016;
Zhang et al., 2020; Al-Azri et al., 2007), although a bloom of *N. scintillans* in southeast Australia only induced ichyotoxicity
when the cell concentration reached 2,000,000 cells $L^{-1}$ (Hallegraeff et al., 2019). For comparison, we observed a maximum
of 32 cells $L^{-1}$ in one microcosm replicate of the olivine treatment.

In comparison to olivine, steel slag seemed to have less potential to affect zooplankton community composition. The
abundance of all groups of phytoplankton, apart from microphytoplankton after day 10, was similar in the slag treatment
and the control through the experiment. This is probably because the amount of slag powder added in the treatment was
much less than the olivine powder resulting in fewer physical particle perturbations to zooplankton. In addition, the
chemistry perturbations such as enhanced alkalinity concentration and various dissolved trace metals, especially Mn, from
the slag powder did not seem to have a notable direct influence on zooplankton abundance over the three-week period.
Even though we did not observe drastic zooplankton abundance changes during the experiment, considering there was
higher microphytoplankton abundance in the slag treatment after day 10, slag powder may benefit some zooplankton
especially those who feed on large phytoplankton on a longer time scale.

**4.4 Dissolved trace metal accumulation in seawater and its environmental implications**
The addition of olivine and slag as OAE source minerals released trace metals into the seawater, predominantly Al, Fe, Ni,
and Cu (olivine) as well as Al, Fe, and Mn (slag). The maximum measured concentrations for dissolved Al, Fe, Ni, Cu, and
Mn were 1190, 500, 80, 30, and 820 nmol $L^{-1}$, respectively. The threshold values for drinking water with health or aesthetic
considerations by the Australian Drinking Water Guidelines for Al, Fe, Ni, Cu, and Mn are 7400, 5360, 340, 15600, and
1800 nmol $L^{-1}$, respectively (NRMMC, 2022). All dissolved trace metal concentrations measured herein are well below
these health and aesthetic threshold values. In natural freshwater sources, the concentrations of Al, Fe, Ni, Cu and Mn are
generally less than 44000, 71400, 510, 156, and 25400 nmol $L^{-1}$ (NRMMC, 2022). Although these natural water data were
primarily derived from rivers and streams, they serve as valuable references for evaluating trace metal release in our
experiment. Thus, mineral additions to the microcosms as simulated here did not increase thresholds for any of the
measured trace metals beyond those that are considered safe for drinking water quality, and they were within the trace
metal concentration range in natural water. However, while these guidelines on drinking water provide a good starting point
on how to qualify what OAE perturbation could be considered "safe" and "unsafe" with regards to trace metals, it must be
recognized that seawater is not drinking water and that critical thresholds may be different in the latter.

The release of trace metals from OAE materials is considered to have relatively strong effects on biology, particularly in
the open ocean where trace metals usually occur in lower concentrations. For example, the oceanic Al, Fe, Ni, and Mn
concentrations are about 2, 0.5, 8, and 0.3 nmol $L^{-1}$ (Bruland and Lohan, 2003; Sohrin and Bruland, 2011). Previous



research on OAE-associated trace metal impacts on individual phytoplankton species grown in laboratory environments
has shown that concentration thresholds beyond which trace metal induces negative effects on fitness likely differ between
species (Guo et al., 2022; Hutchins et al., 2023; Xin et al., 2023). Indeed, our experiment with plankton communities
provides further support that several components of the planktonic food web are affected by OAE. However, our experiment
does not allow determining whether observed effects were primarily invoked by carbonate chemistry, macronutrient (P and
Si), or trace metal perturbations. Thus, dedicated experiments isolating the impact of these factors on plankton will be
required in the future.
**4.5 Particulate trace metal accumulation in seawater and its environmental implications**
The Derwent estuary (where we collected our plankton communities) was highly metal polluted due to industrial practice
(Macleod and Coughanowr, 2019). Both our dissolved and particulate trace metal data indicated high background metal
concentrations, especially for Fe and Zn. Furthermore, the metal:POC ratios found here (Fig. S5) are higher than reported
for open ocean studies or lab cultures. For example, the Fe:POC can vary from 2-136 $\mu$mol mol$^{-1}$ depending on the cultured
phytoplankton species and the environmental dissolved Fe concentration (Kulkarni et al., 2006; Sunda and Huntsman,
1995; King et al., 2012; Boyd et al., 2015). In our results the Fe:POC values ranged from 1200 to 39 000 $\mu$mol mol$^{-1}$, which
may be due to the particulate trace metal richness of the Derwent Estuary (control) and/or the addition of lithogenic particles
(slag and olivine treatment). The presence of abiotic particulate metal sources creates challenges to quantify metal quotas
and then to evaluate metal accumulation effects on biological organisms.
Our study reveals that the added minerals enriched the particulate trace metal pools to various degrees. Consistent with the
dissolved trace metal data, the slag treatment was enriched with particulate Fe and Mn while the olivine treatment was
enriched with particulate Fe and Ni. The enhanced particulate Ni and Mn concentrations were higher than before mineral
additions and the control levels. This is in line with previous research which indicates a positive correlation between
particulate and dissolved trace metal concentrations (Gaulier et al., 2019).
Based on the amounts released through OAE as simulated here (Fig. 4), it appears that Ni and Mn have the highest potential
to cause toxicity in certain marine organisms (Jakimska et al., 2011). These trace metals have the potential to accumulate
in marine organisms over time (bioaccumulation effects), and their increased concentrations in the food chain can lead to
adverse effects on the health and well-being of organisms at higher trophic levels (biomagnification effects). Previous
research has shown the bioaccumulation of Ni on zooplankton (Villagran et al., 2019; El-Metwally et al., 2022), oyster
(Chouvelon et al., 2022), molluscs (Andra Oros, 2010), and fish (Blewett and Wood, 2015); Mn in zooplankton (El-
Metwally et al., 2022), Antarctic bivalve (Husmann et al., 2012), clams (O'mara et al., 2019), and juvenile turbot (Van
Bussel et al., 2014). However, other studies revealed no biomagnification of Ni or Mn in the marine food webs (Sun et al.,
2020; Chouvelon et al., 2019; Campbell et al., 2005; Mathews and Fisher, 2008). Since it usually requires two connected
trophic levels be examined simultaneously (Colaço et al., 2006; Wang, 2002), it is hard to know whether the OAE-related
trace metal addition will be biomagnified. In addition, the bioaccumulation and biomagnification do not necessary result
in toxicity. Therefore, the next step is to investigate whether the enhanced dissolved/particulate trace metal will affect
higher trophic levels to estimate the environmental risks of OAE on other marine organisms.

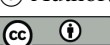



**5 Conclusions**

Our study aimed to assess the environmental impacts of two ground OAE minerals, olivine and steel slag, on coastal plankton communities. Both minerals released alkalinity, leading to an elevation in $pH_T$. However, the addition of steel slag exhibited significantly higher efficiency in elevating alkalinity compared to olivine.

Overall, the application of olivine powder had a noticeable effect on the coastal plankton community. When comparing this impact to the visible perturbation of the plankton community (the seawater turned highly turbid for about 4 days), the impact appears to be modest. Under real-world conditions, dilution through physical mixing with unperturbed water would further mitigate the perturbation. However, when comparing the impact on the plankton community to the limited alkalinity enhancement and $CO_2$ removal potential that was achieved by adding ~1.9 g $L^{-1}$ of olivine powder, it appears to be relatively pronounced. While our experiment was only 3 weeks and olivine powder may slowly release more alkalinity, the short-term response monitored here suggests that the immediate climatic benefit is relatively small compared to a relatively pronounced environmental effect.

In general, the addition of steel slag powder had limited influence on both the phytoplankton and zooplankton community. The major perturbations from slag powder are macronutrients (P and Si) and trace metal (Mn and Fe) additions. Although limited environmental impacts were observed from the slag treatment in our experiment, if slag powders were applied at large scale in the field, these perturbations may circulate in the ocean and influence plankton community well beyond the experimental site. Furthermore, it is essential to consider that the composition of steel slag can vary depending on the source factory (Wang et al., 2011; Proctor et al., 2000), which may affect the efficiency of carbon removal and change the trace metal perturbation. Nevertheless, just based on our experiment, the comparison between the immediate climatic benefit and environmental effect appears to be more favourable as for olivine.

Based on our findings, it can be concluded that steel slag powder exhibited fewer environmental impacts on plankton communities compared to olivine powder, considering its capacity for alkalinity enhancement. The results highlight the importance of carefully assessing the environmental consequences of using specific OAE minerals, particularly when considering their potential effects on plankton communities.

**Data availability.** Data are available in the Institute for Marine and Antarctic Studies (IMAS) data catalogue, University of Tasmania (UTAS) (https://doi.org/10.25959/X6FH-9K15, Guo, J., & Bach, L. (2023).).

**Author contributions.** LTB, RFS, KMS and JAG designed the experiments and JAG carried them out. LTB, RFS and KMS supervised the study. ATT analysed the dissolved/particulate trace metal samples. JAG conducted statistical analyses. JAG prepared the manuscript with contributions from all authors.

**Competing interests.** The contact author has declared that none of the authors has any competing interests.

**Disclaimer.** Publisher's note: Copernicus Publications remains neutral with regard to jurisdictional claims in published maps and institutional affiliations.




**Acknowledgements.** We would like to thank Steve Van Orsouw from Moyne Shire Council, Victoria, Australia for
providing olivine rocks. We also thank Bradley Mansell who provided the Basic Oxygen Slag from Liberty Primary Steel
Whyalla Steelworks in Whyalla, South Australia, Australia. We appreciate Sandrin Feig and Thomas Rodemann for their
support on scanning electron microscopy, particle size measurement, and particulate organic matter. We appreciate the
assistance of Pam Quayle and Axel Durand (IMAS) in the lab, particularly with particulate metal digestions.

**Financial support.** This research has been supported by the Australian Research Council through a Future Fellowship
awarded to Lennart Thomas Bach (project FT200100846), and by the Australian Antarctic Program Partnership
(ASCI000002 to RFS, KMS and JAG). Access to SF-ICP-MS instrumentation was facilitated through ARC LIEF funding
(LE0989539) awarded to ATT.

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
