# Peer review of "Influence of Ocean Alkalinity Enhancement with Olivine or Steel Slag on a Coastal Plankton Community in Tasmania"

_EGUsphere, 2023_

## Author Comment (AC3)

Dear referee,

Thank you for your comments on our manuscript. We appreciate the time and effort that you have dedicated to providing your valuable feedback. Here are our point-by-point responses to these comments and concerns.

**General comment from Reviewer 1:**

This paper presents an interesting and ambitious experiment examining the biological, chemical, and biogeochemical effects of adding olivine ($Mg_2SiO_4$) or steel slag (primarily CaO) to a Tasmanian estuarine plankton community. A particular strength of the experimental design is the use of complex natural plankton communities in relatively large-volume mesocosms, which improves the ecological relevance of the experiment. The biological sampling regime included bacteria and zooplankton in addition to the main focus on the phytoplankton community, and chemical measurements like carbonate chemistry, nutrients and dissolved and particulate trace metals all add additional valuable dimensions to this study. The findings of this incubation experiment will be of great interest to researchers seeking to understand the impacts of proposed OAE mitigation strategies on coastal marine ecosystems.

**Response:** We thank Reviewer 1 for his kind comments.

**Major comment1:**

One important qualification of this study is that Guo et al. must necessarily deal with something of an "apples and oranges" issue with the two alkalinity sources they used. CaO is well known to be a far more concentrated source of alkalinity and dissolves much more readily and rapidly than olivine, as their results also show. This means that it is virtually impossible to make rigorous, quantitative "apples to apples" comparisons between slag and olivine by using equivalent levels of added alkalinity, adding identical weights or volumes of total mineral particulates, adding the same amount of mineral-associated trace metals or nutrients, or by using uniform levels of any other property they have in common to make comparisons. Thus, amounts of each mineral added are necessarily rather arbitrary. This means that direct comparisons between the treatments are highly context-dependent, in that a somewhat different experimental design (i.e., adding more or less of either component) is likely to yield quite different relative outcomes for the chemistry, and perhaps the biology too. This doesn't compromise the value of their experiments, since the general trends in each treatment are still well worth presenting and considering, but it does suggest that interpreting the results in terms of direct quantitative comparisons between the two addition treatments should be done cautiously, and be explicitly qualified in the text.

**Response:** Excellent comment, thank you and we agree with your assessment. The amounts of olivine and slag powder added in the treatments were significantly different resulting in the issue of a quantitative comparison, as you describe. Our original goal was to yield somewhat similar amounts of detectable alkalinity enhancement in the dissolved phase from olivine and slag addition. However, olivine was much less efficient in releasing alkalinity as we expected so that even a 50-fold higher addition of olivine (in mass) did not compensate for this difference. Therefore, our discussion mainly relates the observed environmental effects with the alkalinity enhancement achieved. We added a paragraph section 4.2 of the Discussion pointing out this "apples and oranges" issue to the reader.

**Specific comments:**

**Comment2:** Abstract, lines 22-26: Results for aluminum, manganese and nickel are described here, but many readers will also be looking for the iron results. Perhaps briefly include these in the Abstract too?

**Response:** We added a sentence in the abstract describing dissolved Fe concentrations. "After 21 days, no significant difference was found in dissolved iron concentrations (>100 nmol $L^{-1}$) in different the treatments and the control."

**Comment3**: Methods, Figure 1 and lines 123-126: Using heat belts wrapped around the base of the mesocosms is an interesting and innovative way to control temperatures and set up convective circulation to help keep the plankton community in suspension. Was there a persistent vertical thermal gradient inside the mesocosms? It is also notable that temperatures were fairly variable in some of the mesocosms, with differences of up to 2-2.5C between replicates in some treatments on several days (Fig 2b). Is it possible that this affected replicability of some of the biological parameters?

**Response:** Thanks for pointing this out. The convective circulation methods were established and described in more detail in a recent paper by Ferderer et al., (2022). Convective mixing was monitored using a food dye, and the water colour usually become consistent 20mins after the food dye was added. The time lapse video of the convective mixing test can be accessed online at https://doi.org/10.5446/55861 (Federer, 2021). Yes, this could have affected the replicability of the treatments/control and may have added noise. We added a statement emphasizing that the convective mixing method could have introduced noise in the biological response data (Line 354-356). However, on average there was no statistically significant difference in temperature between control/treatments during the experiment.

**Comment4:** Lines 185-187. The authors should be commended for acknowledging that they are not presenting trace metal or phosphate results from several samples they considered contaminated, a workaround which has often been used in the trace metal literature. However, throwing out 7 of the 36 samples (~23%) is an unusually high proportion of the total. It would perhaps be a good idea to add a small table showing these excluded measurements in the SI so readers can judge the merits of this decision for themselves. Along these lines, it would be good to know what precautions were taken to facilitate trace metal clean water collections, incubations and sampling. Other than some brief description of acid-washing supplies and equipment, no specific precautions are described in the Methods section. I agree with the authors that in situ trace metals in this (quite contaminated) estuary are naturally elevated, and obviously the mineral additions push these even higher, but levels of some easily-contaminated metals like Fe or Zn could still be accidentally significantly increased by sub-optimal experimental protocols.

**Response:** We agree that the proportion of samples we excluded were relatively high compared with other research, and there could be contamination introduced from the sub-optimal experimental protocols. Sampling the microcosm in the temperature control room with potential contaminations coming from the air is the most likely source of contaminations in the dissolved trace metal samples (added in line 191-192). We added a table in the supplementary material (Tabel S1) showing all the raw data including these excluded values. Please note that in previous manuscript, we removed microcosm 5 and 7 on day one because their P concentrations were much higher than other samples excluding the outliers. The P concentrations from these two microcosms measured by ICP-MS were also higher than the P concentrations measured by the spectrometer. However, these values fall into the IQR zone which means they are not outliers determined by IQR methods. So we decide to keep them in the revised manuscript.

**Comment5:** Lines 203-209: The flow cytometer is an excellent way to enumerate single-cell phytoplankton or stained bacteria. However, it doesn't work at all well to count chain-forming, very large or very spiny species like many diatoms and some dinoflagellates, groups which tend to be quite prominent in the coastal ocean. Were any other methods (microscopy, flow cam) used to assess the abundance of these often important groups that are not easily counted with flow cytometry?

**Response:** Unfortunately, we did not use another method to assess the abundance of these large phytoplankton groups. We added a statement noting the potential underrepresented large microeukaryotes abundance in the result (line 471).

**Comment6:** Line 237: Please add a citation to the original paper presenting the oxalate wash cell surface-wash method (Tovar-Sanchez et al. 2003, Marine Chemistry 82).

**Response:** Thanks. We have added the citation (line 245).

**Comment7:** Line 245: The CHN analyses would have yielded numbers for PON as well as the POC discussed here in the Methods and presented later. Are these PON data interesting and potentially worth presenting (possibly in the SI)? The PON values would also allow calculation of changes in whole plankton community C:N ratios, which could be worth examining too. On the subject of ratios, it might be interesting to normalize the BSi values to the POC and the PON to get an idea of the relative degree of silicification of the communities, instead of presenting BSi only as volume-normalized values in Fig 5.

**Response:** Thank you for your comment. We took samples for POC, PON and C:N ratios, but a large part (day 12 to 21) of our POC/PON data were lost due to system failure ( the auto-sampler "ate" these samples). The C:N ratio was calculated for the remaining data, but no significant difference was found between treatments and the control (see Figure 1 below). PON was lower during day 2-5 in the olivine treatment (see Figure 2). In addition, the BSi data during day 2 and 4 from the olivine treatment was removed due to particle influence so we didn't normalize the BSi to POC or PON. Considering these data were not complete and the manuscript is already quite crowded with many plots, we decided not to include PON, C:N or BSi:POC in the manuscript.

[Figure]

Fig. 1. The C:N ratios in microcosms.          Fig. 2. The particulate organic nitrogen concentration

**Comment8:** Line 257: I am concerned about the accuracy of the zooplankton abundance measurements made using the self-made plankton net, which apparently had a diameter of only 1.5cm. Zooplankton tend to be patchily distributed, as they discuss later, and such limited volume sampling is likely be especially problematic for larger, low-abundance groups like larvaceans and copepods. The latter also have issues with active net avoidance that may be quite difficult to deal with using such a small collecting aperture.

**Response:** Thank you for your valuable comment. We apologize for the error in our manuscript regarding the zooplankton net size unit. The correct specifications are as follows: "20cm in height and 15cm in width with a 210µm mesh size," instead of "20mm in height and 15mm in width". We corrected the manuscript accordingly.

**Comment9:** Lines 340-345: This text probably belongs in the Statistics section of the Methods, not the Results.

**Response:** Agreed. We moved this section to the Methods.

**Comment10:** Lines 390-392 and Fig. S3: It is interesting that the olivine released Cu to the seawater, but it is then puzzling that Cu was not reported to be present in the mineral stock used in Table 1.

**Response:** Thank you for your comment. It is likely that Cu was not probably detected by the SEM method used in the analysis of the mineral stocks applied here. This of course does not mean it was not present especially the olivine rock we used in the experiment contains many other particles from the quarry. Nevertheless, to avoid confusion, we change the description of dissolved Cu concentration by deleting the sentence "the olivine treatment released Cu into the seawater" (Line 410).

**Comment11:** Fig. 3: It is surprising that adding only 2 grams of slag to a 53L mesocosm can enrich seawater concentrations of phosphate and silicate to this extent. Although these elevated nutrient additions didn't seem to have much of an effect on the biota in this N-limited experiment, in other regimes where P or Si are scarce this could definitely have a large impact. This issue should be discussed somewhere in the Discussion.

**Response:** We agree. We discussed the implications of P release from slag in section 4.2.1 (line 584-599).

**Comment12**: Lines 385- 405: This section on comparative trace metal releases from both alkalinity sources is a good example of my major general comment above. The relative amounts of metals released from each treatment are clearly a function of the relative amount of each material that was chosen to be added at the beginning: If (for instance) 3g slag had been added instead of 2g, or 50g olivine instead of 100g, the relative release results would likely look quite different. These results as presented are certainly valid, but are very context-dependent in that they only apply to the specific concentrations of each mineral that was added here. One way to deal with this issue would have been to test a range of concentrations of each of these two mineral sources, and examine the data in light of these two gradients. However, I recognize that in a large volume mesocosm experiment this many treatments is usually not practical. I do think some prominent qualifying text in the Discussion is needed to point this important caveat out to readers, though.

**Response:** Thank you, please find our response in relation to your major comment1. We also added a sentence at the beginning of the paragraph emphasizing the amount of minerals added were 50 times different. (Line 405-407)

**Comment13:** Line 419: Fig. 5 needs letters to differentiate the panels.

**Response:** Thanks, letters are mentioned in brackets.

**Comment14:** Lines 458-465: The trends in biovolume for these groups seem to be quite different from those of cell numbers reported in the previous paragraph. For instance, no differences in picoeukaryote, cyanobacteria or cryptophyte biovolumes were observed in any of the treatments

relative to the control, whereas these groups were sometimes significantly higher in the olivine treatment when assessed using cell numbers. Why is this- were there changes in cell diameters and volumes in the treatments? The f.c. results should be able to show this, if so.

**Response:** Thank you for your comment. The data presented in the manuscript are biovolume proportion, which is the proportion of biovolume of a certain phytoplankton type in all phytoplankton types. We have added a sentence in methods describing this calculation (Line 222-225). Because the cell size of different phytoplankton types differs up to 10,000-fold, the changes in cell count of some small phytoplankton types are not obvious in biovolume proportion results. The trends in biovolume and cell counts of each phytoplankton type (Fig. 3) are similar and that's why we did not include the latter in the figure.

[Figure]

Fig.3. The Temporal development of chlorophyll-a concentration (chl-a), BSi, and different eukaryotic and bacterial plankton groups as determined with flow cytometry. (a) chlorophyll-a; (b) BSi; cell concentrations of (c) heterotrophic bacteria, (d) microphytoplankton, (e) nanoeukaryotes2, (f)

nanoeukaryotes1 (g) picoeukaryotes, (h) cyanobacteria, and (i) cryptophytes; biovolume of (j) microphytoplankton, (k) nanoeukaryotes2, (l) nanoeukaryotes1 (m) picoeukaryotes, (n) cyanobacteria, and (o) cryptophytes. The figure data points represent the raw data, and the fitted curve is the generalized additive model. The shaded area represents the 95 % confidence interval.

**Comment15:** Lines 493-512: The much more noisy data for zooplankton than for phytoplankton is likely driven partly by greater patchiness of the former, as suggested here. I suggest this may have also been exacerbated by sampling error from the very small diameter plankton net used to make the collections, as detailed above.

**Response**: Thank you and apologies, there was a typo in our description of the zooplankton net. The net is actually 20cm in height and 15cm in width with a 210μm mesh size while the microcosm is around 510cm tall and 35cm wide. Therefore we think the zooplankton net size is suitable for these microcosms sampling.

**Comment16:** Lines 549-554: I agree, it is hard to draw a direct cause and effect line between higher Fv/Fm and higher abundance.

**Response:** Thank you for your comment.

**Comment17:** Lines 596-609: To my knowledge, no one has shown that Mn or Ni additions can increase photosynthetic efficiency. If so, please cite appropriate references. In addition Mn, like Fe, is typically very abundant in coastal and riverine-influenced waters. I agree that it is puzzling that Fv/Fm increased in the slag and olivine additions despite ambient Fe levels of 100 nM or so, but attributing this response to other metals not known to influence photosynthetic efficiency is quite speculative.

**Response:** Thank you for your feedback. There are some lab experiments indicating the addition of Ni and Mn can enhance the photosynthetic efficiency, like Fv/Fm. These effects are likely species-specific We have cited relevant literature (Pausch et al., 2019; Balaguer et al., 2022; Guo et al., 2022). We agree that it's unknown why the Fv/Fm increased in the mineral addition treatments, and the only theory we can think of is that the coastal phytoplankton community has a higher trace metal requirement than the lab single strain culture. The higher dissolved trace metal concentrations may have elevated the bioactive trace metal concentrations which are easier to be taken up and utilized by the phytoplankton (discussed in line 646-651).

**Comment18:** Lines 617-626: Please see my comments above about patchiness of macro zooplankton and possible sampling artifacts, in view of these questions this paragraph on larvacean trends may be overinterpreting the data a bit.

**Response:** Thank you for your comment. We think part of the confusion probably came from our incorrect description of the zooplankton net in the previous version (we clarified this in Comment8). We agree that the patchy distribution of larvacean, the *Oikopleura* sp., generally brings a large standard error in abundance data, often more than we are used to from phytoplankton data. Nevertheless, we think the interpretation is worth mentioning here because *Oikopleura* is an effective filter feeder, and it is plausible to speculate that the suspended olivine that was highly abundant at the onset of the study slowed down initial growth.

**Comment19:** Lines 719-730: Again, these concluding statements really apply only to the levels of slag and olivine chosen for this particular experiment, and some suggestion of the need for experiments comparing the two under other concentrations and initial conditions is probably needed. One of the reasons CaO sources like slag might have questionable environmental impacts is that they have the

potential to cause extremely rapid and dramatic swings in the carbonate buffer system- for instance, pH increased ~0.5 units over just 4 days in the slag treatments here (Fig. 2a). This of course should drive a correspondingly large and rapid uptake of CO2 from the atmosphere, but could be potentially problematic for some marine organisms. It is reassuring that the plankton community used here was apparently able to accommodate to these major carbonate system swings with little sign of apparent stress. However, this is not necessarily going to be the case for marine metazoans including many invertebrates and fish, which as any aquarist knows are not very tolerant of rapid water chemistry changes. A few words about the need to examine impacts on other trophic levels of the marine food web in the concluding paragraphs would be in order.

**Response:** Thank you for your feedback on this issue. We addressed this comment in our response to comment1. Furthermore, we added text to section 4.2 and in the conclusion that a potential environmental concern of slag may be that it is almost too efficient in that it increases pH too rapidly for some species to acclimate. This must remain a hypothesis for now as our data do not allow firm conclusions. See line 784-791.

**References:**

Balaguer, J., Koch, F., Hassler, C. et al.: Iron and manganese co-limit the growth of two phytoplankton groups dominant at two locations of the Drake Passage. Commun Biol 5, 207, https://doi.org/10.1038/s42003-022-03148-8, 2022.

Guo, J. A., Strzepek, R., Willis, A., Ferderer, A., and Bach, L. T.: Investigating the effect of nickel concentration on phytoplankton growth to assess potential side-effects of ocean alkalinity enhancement, Biogeosciences, 19, 3683-3697, https://doi.org/10.5194/bg-19-3683-2022, 2022.

Pausch, F., Bischof, K., Trimborn, S.: Iron and manganese co-limit growth of the Southern Ocean diatom *Chaetoceros debilis*. PLOS ONE 14, e0221959.. https://doi.org/10.1371/journal.pone.0221959, 2019.
Ferderer, A.: Convective mixing vs no mixing inside two microcosms, TIB [video supplement], https://doi.org/10.5446/55861, 2021.

Ferderer, A., Chase, Z., Kennedy, F., Schulz, K., Bach, L. T.: Assessing the influence of ocean alkalinity enhancement on a coastal phytoplankton community. Biogeosciences, 19, 5375-5399, https://doi.org/10.5194/bg-19-5375-2022, 2022.

---

## Author Response (AR1)

Response Letter

Dear Associate Editor,

Thank you for your comments on our manuscript. We appreciate the time and effort in providing your valuable feedback. Here are our point-by-point responses to these comments.

**General comment:** I thank our two expert reviewers for insightful comments and suggestions, and the authors for their responses. Overall, the manuscript is suitable to be considered for publication upon major revision in line with the reviewer comments, author responses, and my additional comments below.

**Response:**

We thank the associate editor for the generous comment.

**Major comment 1:** In support, and in addition, to the feedback from both reviewers, I recommend that the authors include the following revisions with regard to the experimental treatments:

Abstract - Consider addressing the TA issue in the abstract, as it stands to be a leading question for any reader.

**Response:**

Thank you for pointing it out. We have added a new sentence in the abstract "Olivine and slag powders were of similar grain size, but the amount of added olivine needed to be much higher than the steel slag because less alkalinity is released by the olivine than the steel slag over the 3 weeks experiment." (Line 18).

**Major comment 2:** Methods - Unrelated to the disparate TA outcome, please provide more context for the choice of the intended treatments. The text is currently limited to "The total alkalinity released per amount of mineral powder added was much higher for the slag powder than the olivine powder in our preliminary test trials. So, three microcosms were enriched with 100 g of olivine powder, three microcosms with 2 g of steel slag powder". This does not provide any context for how those levels were selected or what change in TA was targeted and why. It would be worth including the test trial data (L132) as a supporting document.

**Response:**

Thank you, the data we are referring to is part of another project so we cannot show it here (part of it is currently under review). Our goal was to set up two reasonably realistic amounts for different minerals for potential coastal field applications, under consideration of their alkalinity release potential. We have added the following text in the discussion 4.2 to better express how the treatments were designed: "The amount of olivine and slag powder added to the treatments differed significantly (100 g of olivine powder were added while only 2 g of slag powder were added to the 53 L microcosms). Our rationale for these different mass additions was to yield somewhat similar amounts of detectable alkalinity enhancement in the dissolved phase, since we already knew from tests before the experiment that slag elevates alkalinity faster than olivine. However, olivine was less efficient in

releasing alkalinity than we had anticipated so that even a 50-fold higher additions of olivine (in mass) did not compensate for this difference. As such, our experiments are associated with an "apples and oranges issue" in that our perturbation with minerals and associated OAE differs. We argue that an adjusted addition of minerals depending on the alkalinity enhancement rate would be consistent with what OAE practitioners may do under real-world conditions. Presumably, OAE deployments may have to adjust the amounts of minerals to detect alkalinity enhancement in the dissolved phase for verification purposes. Nevertheless, to account for the "apples and oranges issue", the following discussion mainly relates the observed environmental effects with the alkalinity enhancement achieved over the course of the study."

**Major comment 3:** Discussion - The TA outcome from olivine was an unexpected result, but a result nonetheless and it is not discussed. Please include some discussion of this issue. What might have caused the unexpected low dissolution of olivine? Were there methodological differences in the preliminary trial and the experiment? How do the changes in TA observed in the preliminary trials and in the experiment compare to previous olivine experiments or its theoretical dissolution potential? Section 4.1. does not address this nor reference previous work.

**Response:**

Thank your for your comment. The comparative inefficiency of olivine dissolution is indeed interesting. However, we do not have the data available here to make a meaningful contribution as to why the inefficiency occurred. Experiments in our lab and other labs have shown that multiple variables affect dissolution rate. For example, in a study we are currently preparing for publication we found that larger grain sizes, surprisingly, dissolved faster. Also, interaction with sand and the amount of stirring were observed to be a big factor. We think our study adds a lot to the environmental assessment of olivine but provides limited insights into dissolution rates. Thus, we would prefer not to speculate about this here without having the substance for it, especially in light of the publication on olivine dissolution we are currently preparing.

---

## Author Response (AR2)

Response Letter

Dear reviewer,

Thank you for your comments on our manuscript. Here are our point-by-point responses to these comments.

**General comments:**

I enjoyed reading the revised version of the MS by Guo and co-authors and the replies to the comments. I think it's a nice paper that has nicely improved.
I still have a few comments that I will list down here.

**Response:** We thank the review's general comment.

**Major comment 1.** Abstract: I would appreciate if the abstract could be more concise and condensed. The current version lack of appeal while reading it. While I understand it could be challenging, I strongly suggest that the authors make an effort to shorten thinking about the main information they want to convey. Still in the abstract: lines 18-19 can you rephrase it? For example: "Olivine and steel slag powders were of similar grain size. Olivine was added in a higher amount than the steel slag since previous tests evidenced that it would have released less alkalinity over the 3-week experiment".

**Response:** Thanks for your suggestion. We have revised the line 18-19 using your example and shortened the abstract.

**Major comment 2.** Line 356: can you make explicit which day instead of writing "final pH"?

**Response:** We have added the description "the pHT on the day 23" (line 356).

**Major comment 3.** Line 409: "was 50-fold greater than in steel slag (100 g vs 2 g)"

**Response:** We have revised as suggested (now line 403).

**Major comment 4.** Line 461: "treatment"

**Response:** We have added the word accordingly (line 461).

**Major comment 5.** Chapter 4.2: I appreciated the open discussion in chapter 4.2 about the "apples and oranges" issue. Generally speaking, I don't completely get your explanation: you state that you knew already from a previous test that the slag would have elevated alkalinity faster. Since you aimed at reaching a similar TA level through the duration of the experiment for both treatments, why did you add this big mass of olivine from the very beginning, considering all the information achieved in a pre-test?

**Response:** Thank you for your comment. We added the large mass of olivine because the total alkalinity (TA) released by the olivine was not very high (29 $\mu$mol kg$^{-1}$ by 100g of olivine), and this TA elevation is achievable and maybe lower than the real application in the field in the future. We could

have reduced the amount of slag powder added in the slag treatment to achieve a similar TA level as the olivine addition, but since the OAE application may elevate more than 29 µmol kg$^{-1}$ in real application to achieve carbon removal, we decided to use slag powder to assess its environmental impacts on a relatively high TA scenario (around 300 µmol kg$^{-1}$). We agree that the final different TA levels caused some challenges with the comparison of these two materials in our discussion. Thus, our discussion mainly relates the observed environmental effects with the alkalinity enhancement achieved over the course of the study.

**Major comment 6.** Line 570: I would change the word "argue" that is too strong and a bit provocative and I would just say that your study is still relevant since it's consistent with a real-world application of OAE using different materials.

**Response:** We have changed the word "argue" to "note".

**Major comment 7.** Line 585: please change it to "within the alkalinity ranges tested in this study."

**Response:** We have rephased according to your suggestion (now line 578).

**Major comment 8.** Line 594-595: "As such, diatoms are likely to benefit from olivine and slag 594 applications." Can you condense this sentence with the previous one to avoid repetition?

**Response:** Thank you for pointing this out. We have deleted this sentence since it was discussed in the previous one (now line 587-588).

**Major comment 9.** 651-655: These sentences are too speculative...you should dig more into species level to say so. I would be more cautious here if you don't have information at the taxa level or TM cellular requirements of coastal species in the area.

**Response:** We rephased these sentences to make it more precise. "It is possible that these coastal phytoplankton species have higher Fe requirements than those from the open ocean where Fe is limiting (Strzepek and Harrison, 2004). Our findings suggest that Fe perturbations may not only be relevant for low Fe open ocean regions but could also be relevant for coastal ocean locations." (now line 642-645)

**Major comment 10.** Line 690-693: I appreciated that you tried to explain the second bloom of cyanobacteria but your interpretation is not consistent with your data analyses as you mentioned in the text ("The section second bloom of cyanobacteria in olivine is likely to be the results of decreased predators, like Penilia sp. and Oikopleura sp., although the changes in their abundance were not statistically significant between treatments and the control").
**Response:** We apologize for this confusion. The bloom of cyanobacteria was statistically significant but the abundance of Penilia sp. and Oikopleura sp. was not statistically significant possibly due to the limited data points we got and the limitation of GLM on the zooplankton dataset. We have deleted the unprecise description (now line 683).

**Major comment 11.** Due to the "apples and oranges" issue, the sentence in lines 800-801 (from Based to enhancement) should be deleted. Otherwise, you make the same mistake as the previous

version where you compared two different things.
I hope to see this MS published asap! Good luck!!!!

**Response:** Thank you for your comment. We think the sentence as formulated currently is precise and correct because we compare the environmental impact to the alkalinity enhancement potential observed in our study. The slag was >10 more efficient and considering that it would have to have >10 stronger environmental side-effects to have a relatively similar impact than olivine. This was not the case, so that the sentence is correct in our opinion, and we would like to keep it.

---

## Author Response (AR3)

Dear associate editor,

Thank you for your comments again. Here are our point-to-point responses:

Comment 1:
Discussion (Section 4.1). The TA outcome from olivine was an unexpected and it is not discussed. What might have caused the unexpected low dissolution of olivine? Were there methodological differences in the preliminary trial and the experiment? How do the changes in TA observed in the preliminary trials and in the experiment compare to known olivine dissolution kinetics?

Response:

Some preliminary trials were done in bottles and differences may have occurred due to stirring. We note that in the case of the microcosm experiment, olivine was only suspended for a few days while stirring kept it more in suspension over the long timescale. We are currently drafting a paper that is fully focussed on dissolution rates on olivine based on different turbulence scenarios, which will provide the critical information (and focus) on the topic. We don't think that a discussion about data that is not of primary relevance for the findings and not at display can improve the text (but rather focus on the topic in a targeted and comprehensive paper).

To clarify we changed the abstract slightly in order to not raise expectations for too much dissolution rate data and discussion (line 19-20).

Comment 2:
L563-566: Please remove all the speculation as to what olivine OAE practicioners might do. This statement assumes that practitioners are not looking closely at dissolution dynamics in an environment that is ultimately very different from the experimental setup used here.

Response:

Thanks for your comment. We have removed these sentences (line 556-558).

---

## Author Response (AR4)

Dear editors:

Thank you for your comments on the manuscript. Here are our responses:

**Public justification:**
The authors have sufficiently addressed reviewer comments. There was one additional concern that was submitted in private. I consulted with co-editors of this Special Issue about this concern. We agree with the reviewer and so request a minor revision.

We believe there are two places where the conclusions of this paper are too assertive: L34-36 and L759-764. We request a revision to avoid future misinterpretation of this paper that steel slag is a better/safer/more favorable material for OAE than olivine, which this experimental design cannot conclude given that the real-world applications would produce conditions that are different from those tested in this experiment.

In my opinion, the most informative conclusion from this important experimental work to highlight is that steel slag could be an effective material to use for OAE and thus ought to be explored further and, secondarily, the results of olivine ought to be understood in the context of realistic olivine applications and exposure conditions. The manuscript does not require a direct comparison statement in order to be complete. Moreover, I think that adding such a comparison introduces a weakness to the paper, as it invites readers to question the quality and value of that comparison (as exemplified in the review process).

**Response**: Thank you for your comment. We are somewhat hesitant to alter the text based on a private statement. Please let us explain why with a couple of arguments:

1) After carefully analysing our conclusions, we think our text is balanced and fact-based and also within the constraints how these alkalinity sources may need to be applied (see point 2). We acknowledge the time-dependency of the experiment (mention short-term effects) and also that olivine (with the 1.9 g L$^{-1}$ of added material) has potential to increase alkalinity in a longer-term.

2) We are not fully agreeing that our olivine experiments were that unrealistic. It may be necessary to add quite large amounts of olivine to a given water body because otherwise there is no chance to measure its CDR effect. If MRV

frameworks require the measurement of alkalinity release, then the added amounts are on the conservative end because our experiment did not include dilution. To be even more clear about the time-scale dependency and the issues this creates we added a sentence at lines 545 of the revised manuscript.

3) We are concerned that changing the paper based on a private statement lacks transparency as the motivation for the comment is unclear. It could be a bad look for the paper if we alter what we think are data-based conclusions (within the frequently mentioned limitations of the experimental design) towards a very late stage of the process.

Perhaps one way out could be that the private statement is posted on the peer-review forum? That way we could openly address the issue and maintain full transparency.

**Additional private note**:
I recommend updating 'kg' in L544 to 'kg $CO_2$'.

**Response:** Changed as requested.

Kind regards,

Jiaying Guo and Lennart Bach

---

## Author Response (AR5)

Dear editors,

Thank you for posting the statements. Here is our response to your comments.

**Public justification:**
"I appreciated that the authors openly admitted the "apples and oranges" issue. However, in the conclusion, they strongly advocate for choosing steel slag over Olivine, citing its efficiency in CO2 removal and limited environmental impact. I find this statement too assertive, and some rephrasing seems necessary for the following reasons. The reasons are: 1. the authors themselves noted that different steel slags may contain varying trace metals with different impacts on biota. Hence, the final statements seem overly general; 2. the authors themselves highlighted the limited duration of the experiments. It could be possible that Olivine released less TA in this short time and/or that the set-up of the experiments - distinct from a coastal environment with turbulence and strong waves - was not ideal for testing the effective release of alkalinity from an Olivine rock."

As indicated previously, we recommend that a minor revision to L33-35 and L759-763 (of the most recent uploaded version) would be sufficient to address this concern.

**Comments to the author:**
I am recommending the authors make a minor revision of the two statements (or add an additional sentence) to avoid any generalized prescriptions for the use of different alkaline material in OAE projects. There are many considerations that will come into play when choosing what type of material to use for OAE, some (e.g., logistical, LCA related, etc) that are not assessed in this study.

**Response to the public justification:**

Thank you for your careful consideration of this conclusion and constructive comments. We agree with your first point and have emphasized that minerals differ in composure and care must be taken when transplanting our observations to other slags or olivine. We already mentioned that in the conclusion (line 761) but now also added this statement in the first section of the discussion (lines 551-553).

With regard to your second point, we are not agreeing entirely with your argument. It is true that abrasion, potentially induced by wave action, increases olivine dissolution rate (Flipkens et al., 2023). However, such increased dissolution would not only increase alkalinity but also release of other (potentially detrimental) substances such

as specific trace metals. Thus, the higher amount of added material is not the most crucial metric for the comparison with slag but more important appears to be how much material got dissolved.

Our argument is that while increased duration/stirring may have increased alkalinity release, it would also have increased other substances that cause environmental effects. The stoichiometry of release seems to be what matters the most (and not how much material is lying at the bottom of the microcosm). We needed to add substantially more olivine to achieve only a fraction of CDR potential as steel slag and still observed more pronounced environmental effects.  As such our argumentation throughout the text has consistently been to relate the environmental impact to the alkalinity generation. With this concept in mind, we think that our conclusions are fully backed by the data.

Nevertheless, we did the following changes to account for the reviewer's comment on the specificity of the material's impacts:

1) The text in the abstract where we compare slag and olivine CDR potentials and associated environmental effects is now much less general but specified for the specific types of materials used here (line 33-37).
2) We deleted the following statement in the conclusion: "Based on our findings, it can be concluded that steel slag powder exhibited fewer environmental impacts on plankton communities compared to olivine powder relative to its capacity for alkalinity enhancement" (line 766-767).

We hope these adjustments resolve the remaining controversy.

Kind regards,

Jiaying Guo

Flipkens, G., Dujardin, V., Salden, J., T'Jollyn, K., Town, R. M., and Blust, R.: Olivine avoidance behaviour by marine gastropods (*Littorina littorea* L.) and amphipods (*Gammarus locusta* L.) within the context of ocean alkalinity enhancement, Ecotoxicol Environ Saf, 270, 115840, 10.1016/j.ecoenv.2023.115840, 2023.